# OvercookedV2: Rethinking Overcooked for Zero-Shot Coordination

**Tobias Gessler**[*]          **Tin Dizdarevic**          **Anisoara Calinescu**

**Benjamin Ellis**          **Andrei Lupu**          **Jakob N. Foerster**

FLAIR, University of Oxford

## Abstract

AI agents hold the potential to transform everyday life by helping humans achieve their goals. To do this successfully, agents need to be able to coordinate with novel partners without prior interaction, a setting known as zero-shot coordination (ZSC). Overcooked has become one of the most popular benchmarks for evaluating coordination capabilities of AI agents and learning algorithms. In this work, we investigate the origins of ZSC challenges in Overcooked. We introduce a state-augmentation mechanism that mixes states that might be encountered when paired with unknown partners into the training distribution, reducing the out-of-distribution challenge associated with ZSC. Our results show that ZSC failures can largely be attributed to poor state-coverage rather than more sophisticated coordination challenges. The Overcooked environment is therefore not suitable as a ZSC benchmark. To address these shortcomings, we introduce OvercookedV2[1], a new version of the benchmark, which includes asymmetric information and stochasticity, facilitating the creation of interesting ZSC scenarios. To validate OvercookedV2, we demonstrate that mere exhaustive state coverage is insufficient to coordinate well. Finally, we use OvercookedV2 to build a new range of coordination challenges, including ones that require test-time protocol formation, and we demonstrate the need for new coordination algorithms that can adapt online. We hope that OvercookedV2 will help benchmark the next generation of ZSC algorithms and advance collaboration between AI agents and humans.

## 1 Introduction

AI agents are promising to revolutionise our daily lives, from digital assistants (Bai et al., 2022; Team, 2024) to advanced robotics (Leal et al., 2023; Ahn et al., 2024). To achieve their promise, AI agents must be able to collaborate with humans and help them achieve their goals (Kapoor et al., 2024). Doing this successfully requires making decisions where the optimal behaviour depends on the actions of other agents, including humans. Unlike humans, who naturally navigate these situations using theory of mind to reason about the mental states of others (Premack & Woodruff, 1978), current AI agents struggle to adapt their behaviours. Building agents that can coordinate well with humans is therefore one of the fundamental quests of AI (Gupta et al., 2024).

Cooperative multi-agent reinforcement learning (MARL) holds promise for addressing coordination problems but has thus far struggled to handle the complexity of the real world. Most research has focused on learning policies for entire teams, referred to as self-play (SP) (Gupta et al., 2017). Methods such as QMIX (Rashid et al., 2018) and COMA (Foerster et al., 2017) have achieved strong performance on the popular cooperative benchmark SMAC (Samvelyan et al., 2019). While SP agents perform well when interacting with each other, they often struggle when paired with unknown partners since they rely heavily on conventions formed during training. With the success of MARL algorithms, there has been a growing interest in building agents that can coordinate with *novel partners*. One of the main settings this work has focused on is zero-shot coordination (ZSC).

The goal for ZSC, as introduced by Hu et al. (2021b), is to construct an algorithm that allows agents to be trained independently yet coordinate effectively at test time. This approach serves as a proxy

---

[*]Correspondence to `tobias.gessler@cs.ox.ac.uk`

[1]Available in JaxMARL: `https://github.com/FLAIROx/JaxMARL`

for the independent reasoning processes of humans (Hu et al., 2021a). ZSC is commonly evaluated in cross-play (XP), where independently trained models are paired together. Most ZSC research has evaluated on the Hanabi benchmark (Bard et al., 2020), that focuses on reasoning about the beliefs and intentions of other players. This faces the risk of overfitting to the environment Gorsane et al. (2022), and Hanabi's complexity makes training computationally expensive, limits agent interpretability, and is disconnected to real-world tasks. In contrast, the Overcooked benchmark (Carroll et al., 2020), where agents prepare dishes in a restaurant setting, provides easier visualisation and interpretation of agent behaviours in a task inspired by the real world. It has been widely used to study human-AI coordination (Strouse et al., 2022; Zhao et al., 2022; Yu et al., 2023). Despite the benchmark being fully observable, coordination with humans has shown to be challenging.

In this work, we analyse the origin of ZSC difficulties, as measured by the SP-XP gap, in Overcooked. We show that the SP-XP gap is mostly caused by insufficient state-coverage rather than coordination challenges. For example, agents might adopt slightly different movement patterns or timing, causing their partners to encounter unfamiliar states and fail entirely. By employing a state augmentation algorithm during training, we nearly close the SP-XP gap across all layouts. Our algorithm exposes agents to states they might encounter when paired with unknown partners during training, thereby eliminating the out-of-distribution generalisation challenge. State-coverage is one aspect that makes ZSC challenging; however, ZSC may also require learning grounded communication strategies or test-time protocols. Thus, solving state-coverage alone is not equivalent to solving ZSC. As a result, Overcooked is inadequate for benchmarking ZSC algorithms.

To address these shortcomings, we introduce a novel environment, *OvercookedV2*, that requires agents to coordinate for high returns. The environment is implemented as part of the popular JaxMARL framework (Rutherford et al., 2023). We introduce meaningful partial observability by restricting the agents' observations and introducing hidden information sources to ensure that agents cannot simply infer unknown information without interpreting the behaviour of others (Ellis et al., 2023; Foerster et al., 2019). Moreover, we implement distinct communication channels and configurable

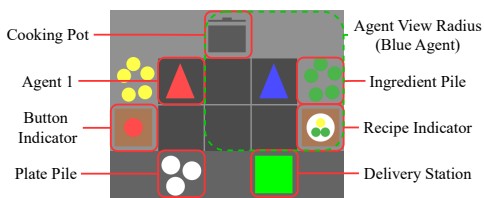

Figure 1: Overview of an OvercookedV2 layout with multiple ingredients, a dynamic recipe indicator, and an agent view radius of one cell.

stochasticity, allowing us to create interesting coordination challenges. This is a significant advantage over Hanabi, which offers only one complex evaluation setting. We provide three handcrafted classes of layouts with specific coordination challenges, along with a user interface for custom layout creation. First, we introduce a temporally extended version of the toy Cat-Dog coordination problem introduced by Hu et al. (2021a). We then present a class of layouts where agents must form protocols at test time to succeed at ZSC, i.e., scenarios where it is not possible to coordinate via a *fixed* policy. Finally, we implement layouts where agents have to rely on the "demoing" of recipes.

To validate OvercookedV2, we demonstrate that the new coordination scenarios cannot be solved by mere state-coverage alone. Moreover, while our results show that ZSC methods such as other-play (Hu et al., 2021b) can improve ZSC performance, all methods struggle with many of the new scenarios, particularly those requiring test-time adaptation. This challenge requires fundamentally new methods capable of adaptive coordination, highlighting an open challenge for ZSC. We hope that OvercookedV2 will help benchmark the next generation of ZSC algorithms and aid in the development of AI systems that can effectively coordinate with both humans and other agents.

## 2 RELATED WORK

**MARL Environments.** In multi-agent setting agents are required either to collaborate or compete with each other. Agents must communicate with each other, develop specialised behaviours, and be robust to the failure of other agents (Gronauer & Diepold, 2022). Standardised environments have facilitated great progress in RL (Bellemare et al., 2013; Brockman et al., 2016). For cooperative MARL, the SMAC Samvelyan et al. (2019), based on StarCraft II, has become popular. Its limited stochasticity and partial observability led to the development of SMACv2 (Ellis et al., 2023). For

evaluating human-AI cooperation and ZSC, the community primarily used the Hanabi (Bard et al., 2020) and Overcooked (Carroll et al., 2020) benchmarks. Most ZSC algorithms have been evaluated in the Hanabi environment (Hu et al., 2021b;a; Lupu et al., 2021), which focuses on reasoning about the beliefs and intentions of other players. However, the complexity of Hanabi makes it difficult to gain insights into agent behaviours. Additionally, the holistic nature of Hanabi makes training computationally expensive and limits the ability to focus on isolated coordination problems.

**Overcooked.** The Overcooked benchmark, introduced by Carroll et al. (2020), is based on the popular video game Overcooked. Players control chefs in a kitchen setting, with the goal of delivering dishes. To deliver a dish, players must work together to complete several high-level tasks. The benchmark attempts to make strategy coordination difficult, in addition to the motion coordination challenge. Players are placed in a grid-world environment containing interactive objects such as cooking pots, onion dispensers, plate dispensers, delivery stations, and empty counters. Players can navigate the environment and interact with these objects. They can perform sequences of interactions to prepare, cook, and deliver dishes. The goal is to deliver a dish which rewards all players equally and removes the dish from the game. The Overcooked benchmark has been widely used to evaluate human-AI coordination algorithms (Strouse et al., 2022; Zhao et al., 2022; Yu et al., 2023). Recent work has explored language model-based agents in Overcooked (Liu et al., 2024; Tan et al., 2024; Li et al., 2024). However, since the benchmark is fully observable without asymmetric information agents might not need to communicate. Moreover, Knott et al. (2021) show that RL agents in Overcooked lack state diversity, which raises questions of their influence on ZSC failures.

## 3 BACKGROUND

**Dec-POMDPs.** A Dec-POMDP is defined as a tuple $(n, \mathcal{S}, \mathcal{A}, T, \mathcal{O}, O, R, \gamma)$ (Oliehoek et al., 2008). Here, $n \in \mathbb{N}$ denotes the number of agents, $\mathcal{S}$ the state space, $\mathcal{A}$ is the action space for each agent, and $\mathcal{O}$ represents an agent's observation space. The transition probability function $T : \mathcal{S} \times \mathcal{A}^n \to \Delta(\mathcal{S})$ dictates the evolution of the environment, the observation function $O : \mathcal{S} \times \mathcal{A}^n \to \Delta(\mathcal{O}^n)$ determines what agents can perceive, and $\mathcal{R} : \mathcal{S} \times \mathcal{A}^n \to \Delta(\mathbb{R})$ is the reward function. Finally, $\gamma \in [0, 1)$ is the discount factor. At every timestep $t$, each agent $i$ selects an action $a_t^i \in \mathcal{A}$. The environment then transitions from state $s_t \in \mathcal{S}$ to $s_{t+1} \in \mathcal{S}$ according to $T(s_t, \mathbf{a})$ with probability $\mathbb{P}(s_{t+1}|s_t, \mathbf{a}_t)$, where $\mathbf{a}_t = (a_t^1, \ldots, a_t^n)$ represents the joint action. A reward $r_t \in \mathbb{R}$ is drawn based on $\mathcal{R}(s_t, \mathbf{a}_t)$, and each agents receives an observation $o_{t+1}^i$ with the joint observation $\mathbf{o}_{t+1} = (o_{t+1}^1, \ldots, o_{t+1}^n)$ sampled from $O(s_{t+1}, \mathbf{a_t})$. The primary objective is to learn policies $\pi_i : (\mathcal{O}^i \times \mathcal{A})^* \to \Delta(\mathcal{A})$ for each agent $i$, which maximises the expected cumulative discounted reward $\mathbb{E}[\sum_{t'=t}^{T} \gamma^{t'-t} r_{t'}]$.

**Partial Observability and Stochasticity.** Partial observability and stochasticity are required for a problem to be truly multi-agent. Ellis et al. (2023) show that when the initial states and transition dynamics of a Dec-POMDP are deterministic, open-loop policies suffice. This means there exists an optimal joint policy where the only required information is the timestep and agent ID, effectively reducing the problem to a predetermined sequence of actions. Open-loop policies render the observation function–and any partial observability it induces–irrelevant. Partial observability only becomes meaningful when the environment is sufficiently stochastic to require closed-loop policies. However, even in stochastic environments, not all partial observability is significant, for example, when it can be inferred from available observations. In fully observable settings, each agent can independently compute all optimal joint policies based solely on the current state, rather than relying on the action-observation history. It can then enact its part of one such joint policy. If both agents do so, they will only fail to coordinate if they choose joint policies that are immediately incompatible.

**Zero-Shot Coordination.** Self-play is the predominant approach for learning in Dec-POMDPs, where agents are trained and evaluated together. For $n$ agents, with $\pi$ denoting the joint policy and $\pi_i$ the $i$-th agent's policy component, self-play optimises $\pi^* = \arg\max_\pi J(\pi_1, \pi_2, \ldots, \pi_n)$. Complex Dec-POMDPs often have multiple maxima, leading to distinct optimal policies that depend on arbitrary conventions which often fail when paired with independently trained agents. However, in many real-world scenarios, agents are required to cooperate with unknown partners or humans at test time. Hu et al. (2021b) formalised the ZSC problem, where the goal is to develop algorithms that enable agents to be trained independently yet coordinate effectively at test time. This can be seen as a proxy for the independent reasoning processes of humans (Hu et al., 2021a) and explicitly

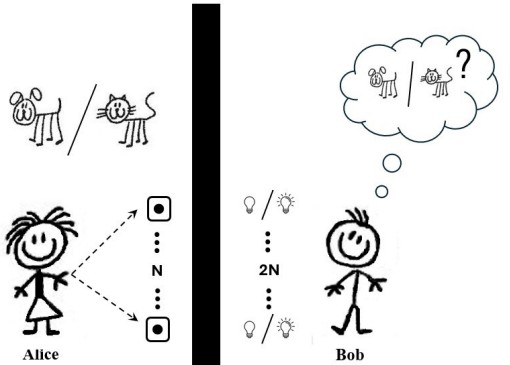
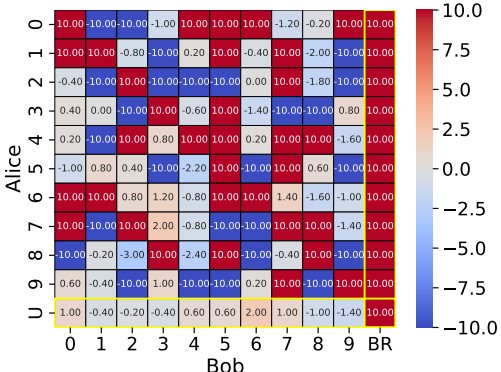

Figure 2: **Button Game.** Cooperative guessing game with many equivalent communication actions. Alice (left) observes a pet, either a cat or a dog, and Bob (right) must guess the pet. Alice must press one of $N$ buttons, activating one of $2N$ light bulbs. The bulb's parity encodes the pet's identity, which Bob observes before guessing the pet. The game does not require coordination as Bob can always guess correctly by looking at the parity of the bulb. Correct/incorrect guesses are rewarded with +/-10 points.

Figure 3: Cross-play matrix for 10 independent SP agents and a best response (BR) to a uniform random agent (U). The game admits a globally optimal strategy for Bob. SP agents succeed in self-play but fail to coordinate with novel partners since they overfit to specific buttons and fail to generalise. The BR agent achieves perfect scores in all pairings. This demonstrates that apparent coordination issues can in some cases be entirely explained by lack of state coverage.

rules out reliance on arbitrary conventions as optimal solutions. A common approach to evaluate ZSC algorithms is cross-play, where independently trained models are paired together. We refer to the difference in performance between self-play and cross-play as the SP-XP-gap. If different training runs of the same method struggle to coordinate, it is unlikely that these agents will succeed with other unknown agents, let alone with human players. Therefore, cross-play serves as a cheap and effective proxy for assessing ZSC performance.

## 4 MOTIVATION: GENERALISATION ACROSS STATES VS COORDINATION

The SP-XP gap observed in ZSC can be caused by either suboptimality, brittle optimal policies, or fundamentally incompatible conventions (what we refer to as a "true coordination challenge"). Sub-optimality results from training runs being unstable or converging to local optima and is not a coordination-specific issue. Making sure that agents generalise across states is generally a goal in RL, but particularly relevant in ZSC, where independently trained agents can adopt similar strategies but with arbitrary variations in execution, such as different movement patterns or timing. While these differences are per-se irrelevant, they can cause partners to encounter unfamiliar states, potentially leading to task failure if agents lack robustness. The Cat-Dog toy problem introduced by Hu et al. (2021a) illustrates a genuine coordination challenge where successful ZSC requires learning a grounded communication strategy which is incompatible with optimal SP policies. In contrast, consider the Button Game (Figure 2, Appendix A.1), an adaptation of the Cat-Dog problem. There is no need for Alice and Bob to establish a specific communication convention, as the information about the pet's identity is always transmitted to Bob, regardless of which button Alice presses. The game does not pose an actual coordination challenge, as agents can achieve a perfect score with unknown partners by generalisation across states alone.

**Experiments in a Toy Environment.** We train 10 independent agent pairs in self-play using independent Q-learning, as well as a BR to a uniform random agent for Alice. The code for this experiment is implemented in a Jupyter notebook and is available for viewing and execution in Google Colab[2] The results are illustrated in Figure 3. Despite the lack of a coordination challenge, agents still fail in a ZSC setting as they overfit to specific button choices and fail in unseen states.

---

[2]https://colab.research.google.com/drive/1GdQ3mRAwMkBxrD-MMgU5Bn7jLDWVPF9a

Table 1: Results of SP and XP scores of 10 PPO agents with distinct seeds in self-play and state-augmented SP for all layouts across 500 episodes (mean $\pm$ standard deviation). The XP score is reported as the mean of all cross-play pairings. The gap represents the difference between SP and XP scores. The SP-XP gap is significantly reduced by training in the State-Augmented setting, demonstrating that much of it was attributable to poor state coverage of SP policies.

| Layout | Standard | | | State-Augmented | | |
|---|---|---|---|---|---|---|
| | **SP** | **XP** | **Gap** | **SP** | **XP** | **Gap** ($\Delta$) |
| *Cramped Room* | $259 \pm 1$ | $257 \pm 2$ | 2 | 259 | $255 \pm 6$ | 4 (2) |
| *Asymm. Advantages* | $278 \pm 1$ | $278 \pm 1$ | 0 | $277 \pm 1$ | $276 \pm 1$ | 1 (1) |
| *Coordination Ring* | $214 \pm 8$ | $96 \pm 72$ | 118 | $269 \pm 44$ | $198 \pm 50$ | 71 (-47) |
| *Forced Coord.* | $199 \pm 1$ | $138 \pm 70$ | 61 | $200 \pm 2$ | $193 \pm 21$ | 8 (-53) |
| *Counter Circuit* | $163 \pm 14$ | $59 \pm 57$ | 104 | $184 \pm 15$ | $140 \pm 38$ | 44 (-60) |

## 5 LIMITATIONS OF OVERCOOKED FOR ZSC

Training RL agents that can coordinate with humans in Overcooked has repeatedly shown to be difficult (Strouse et al., 2022; Yang et al., 2022; Yu et al., 2023). Carroll et al. (2020) provide a web interface[3] that allows users to interact with trained deep RL agents. Interacting with the provided RL agents, including self-play, population-based training, and human-aware PPO agents, reveals significant limitations in their capabilities. Even in the simplest layout, *cramped room*–where all steps required to prepare onion soup can be completed by both players and which primarily presents a basic motion coordination challenge–the agents struggle in certain scenarios. When human players introduce variability, such as placing items like plates on multiple counters, the RL agents fail to make meaningful progress. Although all necessary steps to prepare additional onion soups remain within the agent's capabilities, they often exhibit unproductive behaviours. For instance, they remain stationary in corners or engage in repetitive, non-goal-oriented actions such as repeatedly placing and picking up an onion from a counter. This observation led us to investigate the origins of the SP-XP gap in Overcooked, which we analyse in this section.

### 5.1 EXPERIMENTAL SETUP

To study the origin of the SP-XP gap, we train ten independent PPO (de Witt et al., 2020; Schulman et al., 2017) agents in standard self-play and in self-play with *augmented starting states* for each episode. Specifically, we propose an iterative algorithm to collect these starting states. Before training we rollout $r$ episodes for each policy pairing $(\pi_i, \pi_j)$ and collect the resulting trajectories $\tau_r$. From these trajectories, we select every tenth state and store them in a state buffer $\mathcal{B}$. We then train each policy independently in self-play, with the starting state of each episode sampled uniformly from $\mathcal{B}$. This process is repeated for ten iterations during training with each policy being trained for the same total number of timesteps as in the standard settings. This algorithm exposes agents to states during training, that might have otherwise caused agents to go off distribution during cross-play. However, it is important to note that we still train *exclusively in self-play*; at no point are two different policies trained together, therefore preventing the formation of shared conventions during training. We then evaluate the cross-play performance of all pairings within each of the two populations. Agent implementation details and an overview of the state-augmentation algorithm are provided in Appendix A.2.

### 5.2 RESULTS AND DISCUSSION

Our results show that agents trained in the state-augmented setting perform well across all layouts in cross-play (Table 1). In the Cramped Room and Asymmetric Advantages layouts, even agents trained via standard self-play can collaborate effectively with all other independently trained agents, resulting in an almost negligible SP-XP gap. However, in all other layouts, standard self-play agents show a significant SP-XP gap when paired with unknown partners. By training with augmented

---

[3]https://humancompatibleai.github.io/overcooked-demo

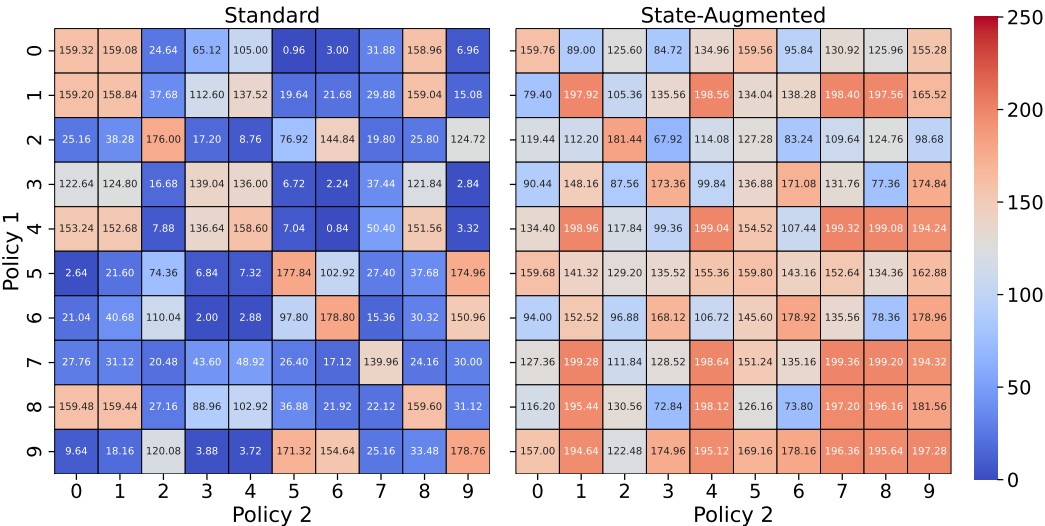

Figure 4: Cross-play matrix for the standard and state-augmented settings in the Counter Circuit layout. Ten agents were independently trained for each setting; each cell $(i, j)$ of the matrix represents the average score across 500 episodes played by the $i$th and $j$th agents. Agents trained in the standard setting achieve high SP scores (diagonal), but most XP pairings perform poorly or fail completely. In the state-augmented setting, SP scores are slightly higher than in the standard setting, and XP scores are dramatically improved, with no total failures.

states, this gap is almost entirely closed in the Forced Coordination layout. Moreover, in the Coordination Ring and Counter Circuit layouts, state-augmented agents also perform well across all cross-play pairings, with no complete failures. Training curves and cross-play matrices for all layouts are provided in Appendix A.

**Results.** In the cramped room and asymmetric advantages layouts, all agents achieve high SP and XP scores, averaging around 260 and 280 respectively. This indicates that there are few high payoff strategies, therefore agents do not encounter unknown situations in XP pairings, which would make them fail. In the Coordination Ring layout, standard agents achieve SP scores of 200-220, but exhibit a significant SP-XP gap. This is likely due to two equally viable strategies that lead to off-distribution failures in cross-play. Agents trained with starting state augmentation improve, achieving an average XP score of 198, with no total failures. In the Forced Coordination layout, standard agents perform well in self-play but some cross-play pairings fail entirely. Starting state augmentation closes the SP-XP gap, with only an eight-point difference. Lastly, in the Counter Circuit layout standard self-play agents perform well, reaching an average total reward of 163 as illustrated in Figure 4. In cross-play, some pairings maintain these high scores, but many achieve only low scores, and some fail entirely. State-augmented agents outperform standard agents, achieving better SP scores and a significantly higher average XP score of 140, demonstrating that the SP-XP gap can be effectively closed by sufficient state-coverage. This layout is larger compared to the others, increasing the number of possible states that agents might encounter when paired with unknown partners.

**Discussion.** The SP-XP gap is largely attributable to poor state coverage, rather than policy inconsistency. By augmenting starting states during training, we reduce the out-of-distribution challenge, leading to significant improvements in cross-play performance across all layouts. In some layouts, even self-play-trained agents collaborated effectively with independently trained agents. This is likely due to a limited number of high-return strategies in those layouts. Most Overcooked layouts have straightforward solutions with few alternative conventions, making ZSC performance more about state-coverage than coordination. Additionally, the fully observable nature of the benchmark does not require agents to communicate asymmetric information. When agents are sufficiently general, they can effectively solve the game effectively with unknown partners. Our findings show that Overcooked is not a particularly hard coordination challenge in ZSC. Instead, the main difficulty lies in state-coverage, suggesting the need for benchmarks that emphasise coordination for high rewards rather than generalisation alone.

# 6 OVERCOOKEDV2

As discussed in the previous section, the original Overcooked environment has significant short-comings. To create a new version of the environment where agents must coordinate to achieve high returns, we propose three major changes: limiting agent observations with a view radius, introducing asymmetric information, and randomising starting positions and directions. By implementing these changes, we introduce meaningful partial observability and increase the stochasticity of the environment, addressing the issues identified earlier. The main objective of the game remains unchanged: agents must prepare and deliver dishes, earning rewards for each successful delivery. For more details on the game Overcooked, refer to Section 2. Our novel environment *OvercookedV2* is implemented as part of the JaxMARL framework (Rutherford et al., 2023), a library that combines JAX implementations of popular MARL environments and algorithms. Implementation details and additional features of OvercookedV2 are available in Appendix B.

## 6.1 ENVIRONMENT DYNAMICS

**Partial Observability.** To introduce partial observability, we limit each agent's observations to a configurable radius around their current position. However, this alone is insufficient to achieve meaningful partial observability. To create scenarios where agents must communicate to maximise rewards, the environment must also include asymmetric information. We therefore extend the environment to feature a configurable number of ingredients, each with its own ingredient dispenser. Now, the environment supports multiple distinct recipes, where each recipe consists of a multiset of three ingredients. We extend the game by adding a recipe to the global state, denoted as $R$. At the start of each episode, a recipe $R$ is sampled from a set of possible recipes $\mathcal{R}$, with an option to resample after each successful delivery. Asymmetric information is introduced in a layout through a recipe indicator block, which displays the recipe that needs to be cooked and delivered. This indicator is only visible to agents within its view radius. A reward is given if the delivered dish matches the indicator, with an optional parameter to impose a negative reward for incorrect deliveries.

**Stochasticity.** Our environment introduces additional sources of stochasticity, most notably through the recipe sampling process, as detailed in the previous paragraph. The probability $P(R)$ of selecting a recipe from $\mathcal{R}$ is equal for all recipes. We also provide varying levels of stochasticity for agent initialisation. In the default case, agents are placed at predetermined positions, all facing upwards, as in the original environment. Alternatively, starting positions and orientations can be randomised, sampled uniformly from all reachable positions for each agent. This is particularly relevant in layouts with separate rooms that restrict agents' access to certain items and information. All interactions with objects, such as pots and counters, remain deterministic.

**Communication Channels.** To collaborate effectively in the presence of asymmetric information, agents must communicate. In Overcooked, agents can exchange arbitrary signals by placing items on counters within another agent's view radius or by observing each other's movements. Arbitrary communication allows agents to exchange signals that hold no inherent meaning and can be used to establish communication protocols. In contrast, grounded communication provides agents with verifiable information, independent of the actor's intentions. To enable grounded communication, we introduce a new block that reveals the recipe when an agent interacts with it, remaining visible for a configurable number of timesteps. Pressing the button incurs a configurable negative reward. Additionally, we implement an option to provide agents with a signal for a successful delivery.

## 6.2 LAYOUTS

The grid-world nature of Overcooked allows the environment to be configured with different layouts to investigate different aspects of coordination. To design a new layout, one can simply provide an ASCII string that describes the layout configuration, along with an optional list of possible recipes. A detailed explanation and example for the interface is provided in Appendix B.5.2. We provide pre-configured layouts for all scenarios present in the original Overcooked benchmark and adapted versions of these layouts with multiple ingredients. Additionally, we introduce three classes of handcrafted layout configurations that pose interesting coordination challenges. Our layouts feature non-obvious ZSC solutions, which cannot be solved effectively with arbitrary conventions. Each class is described in the following, and an overview of the layouts is shown in Figure 5.

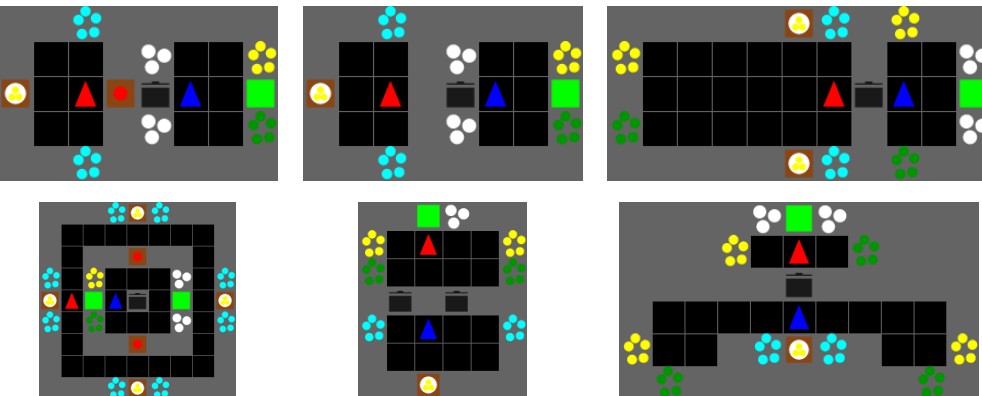

Figure 5: Handcrafted coordination challenges. We provide three classes of layouts, each with a simple (top row) and a more complex (bottom row) configuration. From the left to right: *Grounded Coordination Simple/Ring* layouts present a temporally extended version of the toy Cat-Dog coordination problem introduced by Hu et al. (2021a). *Test-Time Protocol Formation Simple/Wide* layouts require agents to form protocols at test-time, based on feedback they receive after delivery. *Demo Cook Simple/Wide* layouts require agents to rely on the meaning of the other agents' actions.

**Grounded Coordination.** We temporally extend the Cat-Dog coordination problem (Hu et al., 2021a) into a layout. Only one agent can observe to recipe indicator and the other one must prepare the dishes. The agent observing the indicator can opt to use arbitrary signals in the form of placing ingredients on counters or performing certain movements in the view radius of the other agent. Moreover, we include a button indicator as a grounded communication channel, where the agent can press a button and, for a cost, reveal the current recipe. For agents to succeed in ZSC they need to rely on grounded communication and ideally form protocols at test-time. In self-play agents would adopt arbitrary conventions to circumvent paying the cost of pressing the button, which will fail in a ZSC setting. We expect OBL to perform well by learning to communicate using the button indicator.

**Test-Time Protocol Formation.** These layouts again include asymmetric information, requiring agents to communicate. While no grounded coordination channel is available in this layout, agents receive feedback when a delivery is successful. To play with unknown partners, agents can use the grounded delivery feedback to form arbitrary protocols at test time. Again, self-play agents will establish arbitrary conventions during training to solve the communication effectively. Instead of learning fixed conventions during training, agents have to adapt and coordinate on communication protocols at test time to succeed at ZSC in these layouts. We also expect OBL to struggle in this layout, as agents do not have a grounded way to communicate. The formation of protocols at test time requires novel ZSC methods capable of adapting online to other agents.

**Demo Cook.** In the demo cook layouts, agents operate under asymmetric information with no grounded communication or feedback. The layouts require implicit communication through actions, as the choice of ingredient placed in the pot conveys information about the recipe. For ZSC, agents must interpret these actions and optimally use the common understanding to form test-time protocols. Communication through implicit actions is common for humans; however, it might present a significant challenge for AI agents.

## 7 OVERCOOKED V2 EXPERIMENTS

In this section, we present the results of several experiments conducted in the OvercookedV2 environment. We introduce a new model architecture designed for learning in the complex, partially observable coordination scenarios created by OvercookedV2 and demonstrate its strong performance. We then compare the ZSC performance of various algorithms across different layouts in our new environment. Our results show that OvercookedV2 presents hard coordination challenges and that good ZSC performance is not merely an issue of state coverage.

Table 2: Results of SP and XP scores for 10 seeds with different methods in multiple layouts across 500 episodes (mean $\pm$ standard deviation). The XP score is reported as the mean of all possible pairings. The Gap represents the difference between SP and XP. Both self-play and state augmented agents perform well in SP, but score dramatically lower in XP for most layouts. Other-Play SP scores are slightly lower than self-play agents scores, however XP scores improve in most layouts. All methods struggle in cross-play demonstrating that the new scenarios provide a hard ZSC challenge.

| Layout | Method | SP | XP | Gap |
|---|---|---|---|---|
| *Grounded Coord. Simple* | Self-Play | $154 \pm 10$ | $-55 \pm 124$ | $210 \pm 51$ |
| | State-Augmented | $156 \pm 29$ | $-17 \pm 115$ | $176 \pm 69$ |
| | Other-Play | $122 \pm 24$ | $4 \pm 73$ | $116 \pm 47$ |
| | Fictitious Co-Play | $64 \pm 58$ | $46 \pm 47$ | $22 \pm 45$ |
| *Grounded Coord. Ring* | Self-Play | $164 \pm 10$ | $-12 \pm 75$ | $175 \pm 28$ |
| | State-Augmented | $165 \pm 7$ | $-16 \pm 65$ | $183 \pm 26$ |
| | Other-Play | $121 \pm 36$ | $-9 \pm 21$ | $131 \pm 37$ |
| | Fictitious Co-Play | $86 \pm 34$ | $6 \pm 46$ | $79 \pm 29$ |
| *Test Time Simple* | Self-Play | $145 \pm 22$ | $-81 \pm 99$ | $220 \pm 26$ |
| | State-Augmented | $161 \pm 18$ | $-55 \pm 103$ | $230 \pm 53$ |
| | Other-Play | $121 \pm 37$ | $-3 \pm 51$ | $131 \pm 50$ |
| | Fictitious Co-Play | $35 \pm 44$ | $6 \pm 29$ | $25 \pm 47$ |
| *Test Time Wide* | Self-Play | $194 \pm 12$ | $-30 \pm 96$ | $220 \pm 31$ |
| | State-Augmented | $137 \pm 82$ | $-12 \pm 72$ | $142 \pm 81$ |
| | Other-Play | $175 \pm 54$ | $-11 \pm 42$ | $193 \pm 61$ |
| | Fictitious Co-Play | $95 \pm 56$ | $23 \pm 40$ | $68 \pm 50$ |
| *Demo Cook Simple* | Self-Play | $172 \pm 19$ | $36 \pm 82$ | $138 \pm 48$ |
| | State-Augmented | $170 \pm 16$ | $112 \pm 50$ | $54 \pm 36$ |
| | Other-Play | $149 \pm 25$ | $47 \pm 48$ | $96 \pm 19$ |
| | Fictitious Co-Play | $22 \pm 48$ | $1 \pm 25$ | $25 \pm 48$ |
| *Demo Cook Wide* | Self-Play | $195 \pm 4$ | $40 \pm 61$ | $140 \pm 13$ |
| | State-Augmented | $192 \pm 7$ | $60 \pm 60$ | $138 \pm 23$ |
| | Other-Play | $162 \pm 23$ | $62 \pm 53$ | $104 \pm 45$ |
| | Fictitious Co-Play | $45 \pm 43$ | $13 \pm 28$ | $31 \pm 34$ |

**Experiment Setup.** We find that models using a similar architecture as proposed by Carroll et al. (2020) did not learn effectively under the additional complexity in OvercookedV2. We modify the network mainly by adding three 1x1 convolutions to transform the sparse observations of the environment. In addition, we add a recurrent layer to the network. Furthermore, we implement other-play a ZSC algorithm to prevent coordinated symmetry breaking. A limitation of other-play is that the set of symmetries must be known beforehand. This is particularly relevant in Overcooked, where we cannot define a universal set of symmetries for the entire environment; instead, symmetries depend on the specific layout. In our layouts we permute the recipe ingredients while keeping all other ingredients fixed. Moreover, we evaluate the population based ZSC algorithm FCP (Strouse et al., 2022), where we learn an agent as the best response to a population of self-play agents and their past checkpoints. Implementation details are available in Appendix C.1.

**Results Adapted Layouts.** Agents trained with our new architecture achieve high scores in partially observable versions of all original overcooked layouts. They efficiently use of both pots present in the layouts to cook multiple dishes simultaneously; a behaviour that previous policies struggled with. Furthermore, we evaluate agents in adapted version of these layouts with multiple ingredients. Agents are able to maintain strong performance in these layouts even while conditioning on the recipe indicator. Detailed results can be found in Appendix C.2.

**Results Coordination Challenges.** All methods struggle to perform well in XP across all layouts, as shown in Table 2. In the Grounded Coordination layouts, self-play and state-augmented agents achieve similar SP scores. However, in cross-play, state-augmented agents show an improvement for the simpler version of the layout, whereas in the ring layout, XP scores are similar. In the Test-Time

Protocol layouts, agents also experience slight improvements with state augmentation. Other-play agents achieve slightly lower SP scores but generally improve in XP across most layouts. In the Demo Cook layouts, agents perform better in XP compared to the other layouts, though the SP-XP gap remains significant. FCP SP scores are generally lower than those of self-play agents. However, even in cross-play, FCP agents achieve a positive average score, though these scores remain much lower than the SP scores of self-play agents. None of the methods successfully close this gap.

**Discussion.** When inspecting the animation of episodes it is clear that self-play agents rely on arbitrary conventions such as placing an ingredient on the middle counter to communicate. When paired with an unknown partner these conventions do not hold, and agents exhibit unproductive behaviours such as stalling completely or constantly delivering an incorrect recipe. We provide a collection of episode visualizations to demonstrate these behaviours[4]. While other-play helps improve ZSC performance in most layouts, the scores are still far worse than in self-play. This is particularly evident in the Test-Time Protocol layouts, where agents experience a large SP-XP gap. We also observe that FCP agents score significantly lower than self-play agents, even in SP. This could be due to SP eval being out of distribution for FCP agents. Moreover, we use a relatively small population size, FCP could benefit from larger populations. Overall, OvercookedV2 presents a challenging ZSC environment that none of the evaluated methods solve successfully.

## 8 CONCLUSION

Our work revealed significant, largely unexplored, limitations of the Overcooked benchmark, which has been widely used to benchmark human-AI coordination. The finding that Overcooked does not present a meaningful ZSC challenge raises concerns about previous methods evaluated solely in this environment. Evaluation of ZSC algorithms in a single environment, i.e., Hanabi, leading to community-wide overfitting to this environment, is a common issue in MARL research (Gorsane et al., 2022). To address this, we introduced OvercookedV2, providing a more challenging and flexible coordination benchmark. The benchmark, despite its complexity, cannot represent all ZSC challenges; evaluation should therefore not be limited to this benchmark. We provide an initial set of layouts featuring handcrafted coordination scenarios and welcome contributions from the community to OvercookedV2. We anticipate that OvercookedV2 can help benchmark the next generation of ZSC algorithms and aid in the development of collaborative AI systems that can coordinate with both humans and other AI agents.

### ACKNOWLEDGEMENTS

AC acknowledges funding from a UKRI AI World Leading Researcher Fellowship (Grant EP/W002949/1) and from a JPMC Research Award. JF is partially funded by the UKI grant EP/Y028481/1 (originally selected for funding by the ERC). JF is also supported by the JPMC Research Award and the Amazon Research Award.

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

---

**Algorithm 1** State-Augmented Self-Play Algorithm

---

1: **Input:** Number of policies $P$, number of iterations $I$, number of rollouts $R$
2: Initialise each policy $\pi_i$ for $i = 1, \ldots, P$
3: **for** each iteration $k = 1$ to $I$ **do**
4:     Initialise an empty state buffer $\mathcal{B}$
5:     **for** each pair of policies $(\pi_i, \pi_j)$ **do**
6:         **for** each rollout $r = 1$ to $R$ **do**
7:             Collect trajectory $\tau_r$ for the policy pair $(\pi_i, \pi_j)$
8:             Store every tenth state from $\tau_r$ in the buffer $\mathcal{B}$
9:         **end for**
10:     **end for**
11:     **for** each policy $\pi_i$ **do**
12:         Configure the environment to select a random state from the buffer $\mathcal{B}$ as the initial state on each reset
13:         Perform self-play for $N$ timesteps
14:     **end for**
15: **end for**

---

## A    ADDITIONAL MATERIAL FOR LIMITATIONS OF OVERCOOKED

### A.1    BUTTON GAME

In the Button Game, illustrated in Figure 2, Alice observes a pet, which can be either a cat or a dog, and Bob must guess the identity of the pet. We define $pet\_bit = 0$ if the pet is a cat and 1 if it is a dog. After observing the pet, Alice selects an action by pressing one of $N$ available buttons. Pressing a button activates one of $2N$ light bulbs that Bob can observe. The light bulb at position $2a + pet\_bit$ will light up, where $a$ is the button Alice pressed. Bob must then guess whether Alice saw a cat or a dog. Note that there is no need for Alice and Bob to establish a specific communication convention. The parity of the light bulb encodes the pet's identity regardless of which button Alice presses. Both agents receive a reward of +10 points for a correct guess and -10 points for an incorrect guess.

### A.2    OVERCOOKED EXPERIMENTAL SETUP

We perform all experiments in the JaxMARL implementation of the OvercookedV2 environment. Experiment code is available at `https://github.com/overcookedv2/experiments`. We configure the environment to be fully observable and match the environment dynamics of the original implementations. The observations differ to the original implementation since we have dedicated layers for newly introduced objects such as the recipe indicator. However, these layers in the observation grid will simply be zero when objects are not used in the layout. For more details on the environment implementations refer to Section 6.

We train agents using independent PPO, with parameters shared among all agents. The training runs are parallelised across multiple environment instances simultaneously. Our network architecture is inspired by previous work in Overcooked by Carroll et al. (2020). We employ a feed-forward network, parameterised by three 2D convolutional layers with 32 channels each, and kernel sizes of 5x5, 3x3, and 3x3, respectively. We pad our inputs using zeros and employ ReLU as the activation function after each convolutional layer. The output of the convolutional encoder is flattened and passed through a dense layer with 64 neurons. We apply LayerNorm (Ba et al., 2016) to this output, which is then fed into another dense layer with 64 neurons. The final output is used in the actor-network to parameterise a categorical distribution and by the critic to produce a value estimate through a projection layer. We use the Adam optimiser (Kingma & Ba, 2017) to update the network parameters. Additionally, we apply a learning rate warmup at the beginning of training, followed by a cosine learning rate decay schedule. This helps to stabilise training and improve performance (Kalra & Barkeshli, 2024). The same hyperparameters are used for both the standard and state-augmented settings; an overview is provided in Appendix 3. Our experiments were conducted on a server equipped with 8 NVIDIA A40 GPUs with 48GB of memory and an AMD EPYC 7513 32-Core Processor. The models were trained using JAX (Bradbury et al., 2018) and FLAX (Heek et al., 2023).

## A.3    ADDITIONAL FIGURES

This appendix provides supplementary figures that offer deeper insights into the experiments and support results discussed in the main text. We first provide additional figures for the experiments conducted for analysing the limitations of Overcooked and then the experiments conducted in OvercookedV2.

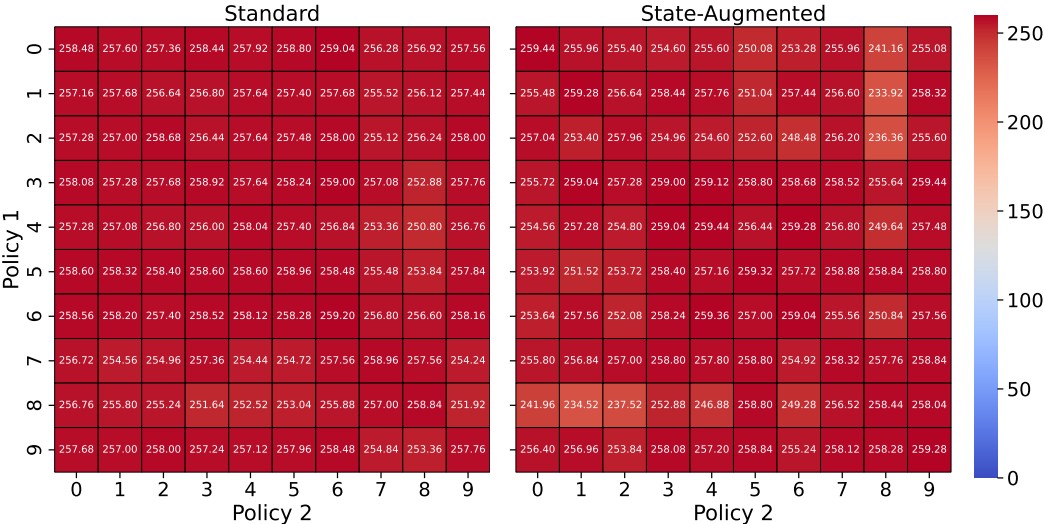

Figure 6: Cross-play matrix for the standard and state-augmented settings in the Cramped Room layout. Ten agents were independently trained for each setting; each cell $(i, j)$ of the matrix represents the average score across $500$ episodes played by the $i$th and $j$th agents. Agents in both settings achieve nearly identical scores in self-play and cross-play, resulting in an SP-XP gap that is nearly zero.

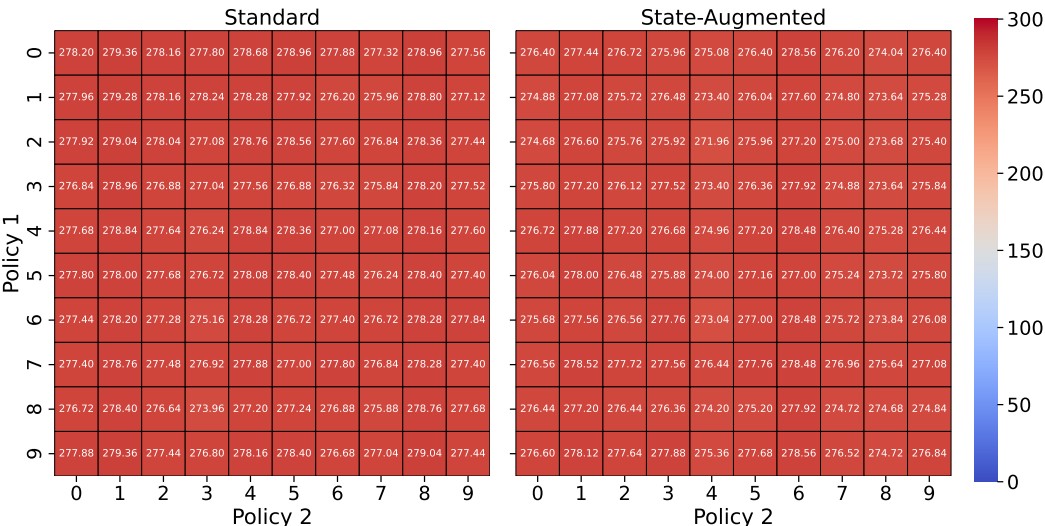

Figure 7: Cross-play matrix for the standard and state-augmented settings in the Asymmetric Advantages layout. Ten agents were independently trained for each setting; each cell $(i, j)$ of the matrix represents the average score across $500$ episodes played by the $i$th and $j$th agents. Agents in both settings achieve nearly identical scores in self-play and cross-play, resulting in an SP-XP gap that is nearly zero.

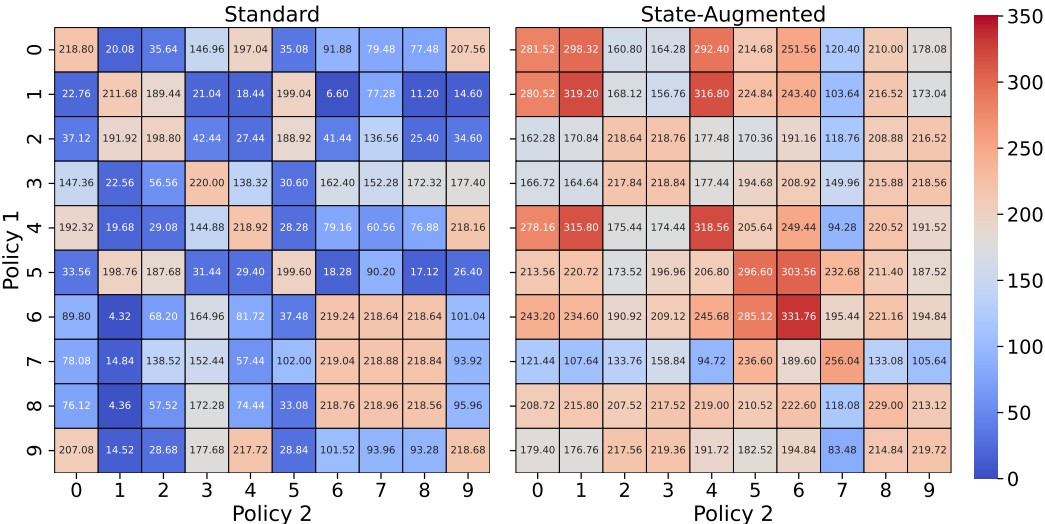

Figure 8: Cross-play matrix for the standard and state-augmented settings in the Coordination Ring layout. Ten agents were independently trained for each setting; each cell $(i, j)$ of the matrix represents the average score across 500 episodes played by the $i$th and $j$th agents. All standard self-play agents achieve similar SP scores of around 200-220 points. While some pairings perform as well in cross-play as in self-play, many fail to deliver effectively, resulting in only a few successful deliveries. In contrast, in the state-augmented setting, half of the agents achieve similar self-play scores, with some even discovering more efficient strategies, yielding scores around 300 points. Importantly, all policy pairings in cross-play perform well, with no complete failures.

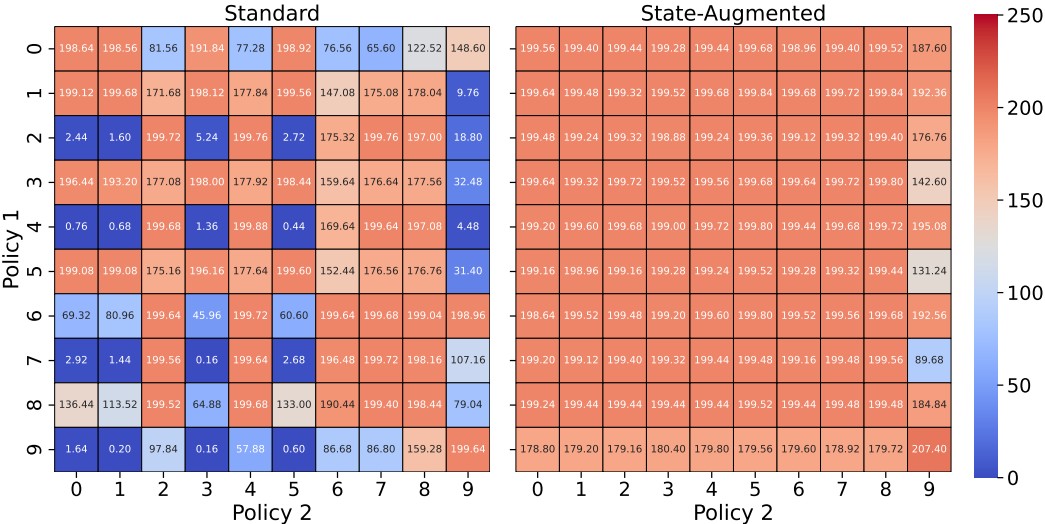

Figure 9: Cross-play matrix for the standard and state-augmented settings in the Forced Coordination layout. Ten agents were independently trained for each setting; each cell $(i, j)$ of the matrix represents the average score across 500 episodes played by the $i$th and $j$th agents. All agents achieve similar self-play scores of around 200 points. While some pairings perform equally well in cross-play as in self-play, others fail entirely, with some not completing even a single delivery on average. In contrast, agents in the state-augmented setting also achieve similar self-play scores, and all policy pairings in cross-play perform well, with no complete failures.

| Plated | Cooked | Ingredient 1 | Ingredient 2 | Ingredient 3 | $\cdots$ |
|--------|--------|--------------|--------------|--------------|----------|
| 1 bit | 1 bit | 2 bits | 2 bits | 2 bits | $\cdots$ |

Figure 10: Bit-encoding scheme for ingredients, plates and dishes. The first bit indicates whether the item is cooked (0: Not Cooked, 1: Cooked), and the second bit indicates whether it is plated (0: Not Plated, 1: Plated). Each ingredient is represented by 2 bits, encoding 0 to 3 units of the ingredient (00: 0 units, 01: 1 unit, 10: 2 units, 11: 3 units).

## B  OVERCOOKED V2 ENVIRONMENT DETAILS

In this section we provide additional details for the environment dynamics, implementation and features of OvercookedV2. While we reuse some of the interaction logic from the existing Overcooked implementation in JaxMARL, the need for multiple ingredients and a configurable number of players requires a new state representation and significant reworking of the interaction logic. In the following, we detail the dynamics of the game, explain the structure of observations, and show the compatibility to the original Overcooked environment. Finally, we provide more details for the provided layouts as well as a guide for the layout creation interface.

### B.1  STATE REPRESENTATIONS

The state of the game consists of the game grid $G$, time $t$, recipe $R$, and agents $\mathcal{A} = \{a_1, a_2, \ldots, a_n\}$. One of the main challenges is allowing for a dynamic number of ingredients for each cell of the grid and in agent inventories. Since JAX requires static shapes for arrays, we developed a bit-encoding scheme to represent ingredients, as illustrated in Figure 10. In this scheme, the first two bits serve as flags to indicate whether an item has been cooked or plated. The subsequent bits represent the quantity of each ingredient in the item. We allocate two bits per ingredient, which is sufficient to encode up to three instances of the same ingredient. Given that our recipes are fixed at three ingredients, this encoding can represent all possible combinations of (partial) dishes. This encoding scheme is applied to all representations of ingredients throughout the game.

The game grid is represented as a three-dimensional grid with dimensions width $\times$ height $\times$ 3. The first layer contains static objects, which include: empty, wall, goal, pot, recipe indicator, button recipe indicator, plate pile, and ingredient piles. The second layer holds the encoded ingredients at each cell. The third layer stores additional information such as pot and button timers. Agents are represented by their position, direction, and inventory.

### B.2  ENVIRONMENT DYNAMICS

One step in the environment consists of three sub-steps, as illustrated in Figure 11. First, we attempt to move all agents to their new positions, if possible. Initially, the new positions are computed without considering collisions. If an agent attempts to move to a location occupied by an object, the move action is not executed; the position remains unchanged, although the agent's direction is updated regardless of whether movement occurs. After computing all movements, we resolve collisions iteratively. During each iteration, we identify all collisions on the grid, mark the involved agents, and reset their positions to their previous locations. This procedure continues until no further collisions are detected. Iterative resolution is necessary because, with more than two agents, resolving one collision might cause another collision with a different agent. Finally, we ensure that no pair of agents has swapped positions, as this is not a valid move; any agents that have swapped positions are reset to their original locations.

Next, we process all agent interactions sequentially, starting with the first agent. If an interaction is not allowed, the action results in a no-op. The outcome of an agent's interaction depends on both the agent's inventory and the object the agent is facing. For example, if an agent is facing an ingredient dispenser and its inventory is empty, the agent picks up one unit of the ingredient. For

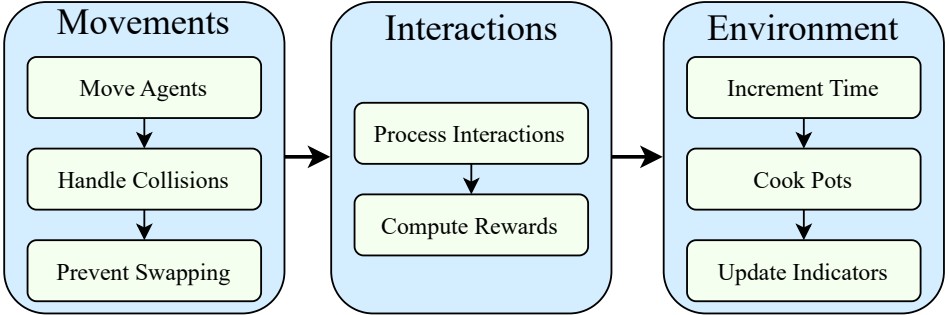

Figure 11: Illustration of the three sub-steps within a single environment step. First, agents are moved to their new positions. Then, interactions with the environment are processed. Finally, the global environment state is updated.

each correct delivery, agents receive a reward of 20 points. If negative rewards are enabled, agents receive -20 points for an incorrect delivery. Furthermore, we provide shaped rewards similar to those in the original Overcooked environment. Rewards in Overcooked can be sparse since preparing and delivering a dish requires many steps. We provide shaped rewards for specific actions, such as placing an ingredient into the pot, picking up a plate while a dish is cooking, and picking up a dish. However, with the addition of multiple recipes, these actions may not always be beneficial, for example adding ingredients to the pot that do not match the current recipe $R$. We only give these rewards when the actions are aligned with the current recipe.

Finally, we perform updates required at each timestep that are unrelated to agent interactions. This includes incrementing the global timer $t$, updating cooking pots, and decrementing button timers. The timers for pots and button recipe indicators, stored in the extra info layer, are decremented by one. We provide a parameter to configure whether agents must interact with a pot to start the cooking process; if this parameter is not set, cooking starts automatically when the pot is full. Once a pot finishes cooking, the cooked flag is added to the ingredients in the pot.

### B.3 Observations

The observations are structured as a *width* × *height* × *num_channels* tensor. If a view radius is configured, agents only observe the partial state within this radius, with any areas outside the game grid being padded with zeros. Most of the layers for each cell are one-hot encoded. Each cell in the grid includes the following components: a one-hot encoding of static objects (e.g., wall, pot, etc.), and an ingredient encoding for items on counters and in pots. Agents are represented by an indicator of their position, a one-hot encoding of the direction they are facing at their respective cell, and an encoding of their inventory. Separate channels are used to encode all other agents and the agent receiving the observation. We also include a separate layer for encoding the ingredients in recipe indicators. All ingredients are encoded using the same scheme: the first two channels indicate whether the item is cooked and/or plated, while the following channels represent the number of instances of each ingredient present in the cell. This encoding scheme is consistently applied across all ingredient layers in the observation.

### B.4 Episode Visualisation

We provide a visualisation engine that renders a game state into an image. The visualisation logic has been adapted from the JaxMARL implementation for the original Overcooked environment. The implementation has been extended to support the visualisation of all new objects, as well as multiple ingredients and recipes. Additionally, we have rewritten the implementation to enable JIT compilation of the entire rendering pipeline, which significantly speeds up the rendering process. This optimisation allows for vectorised mapping over entire state sequences and even across batches of sequences. We offer a user-friendly interface to render entire episodes into GIFs for easy inspection.

Figure 12: Illustration of three different layouts adapted from the original Overcooked environment to include multiple recipes. From left to right: Cramped Room V2, Asymmetric Advantages with recipe indicator on the right and Two Rooms forced coordination layout.

## B.5 LAYOUTS

In this subsection we provide additional details for the layouts included with OvercookedV2. Furthermore, we introduce the layout creation interface in detail, which allows researchers to build layouts simply by specifying an ASCI string and a set of possible recipes.

### B.5.1 ADAPTED LAYOUTS.

We provide pre-configured layouts for all five of the layouts introduced by the original Overcooked benchmark: Cramped Room, Asymmetric Advantages, Coordination Ring, Forced Coordination, and Counter Circuit. By setting no agent view radius, it is possible to replicate the environment dynamics of the original benchmark. Additionally, we offer several simple layouts that include multiple recipes, some of which are minor adaptations of the original Overcooked environments, as illustrated in Figure 12. All these layouts feature two ingredients – onions (yellow) and broccoli (green) – and allow for all possible recipe combinations, resulting in a total of four distinct recipes. We provide a variant of the Cramped Room layout, which has the same room structure as the original but adds a recipe indicator and an additional ingredient. Additionally, we offer adaptations of the Asymmetric Advantages layout, which maintain the original challenge but now include multiple recipes. We provide different variants of this layout, with the recipe indicator positioned on the left, centre, or right. In a partially observable setting, one agent must wait for the other to 'demo' the recipe. We also introduce a layout featuring two rooms where agents must collaborate to cook dishes. In this layout, the agent on the right has access to ingredients, plates, a recipe indicator, and the delivery station, while the agent on the left is the only one with a pot to cook. This setup forces agents to work together, similar to the Forced Coordination layout.

### B.5.2 LAYOUT CREATION INTERFACE

In addition, we offer an easy-to-use interface for researchers to create their own custom layouts presented in Figure 13. To design a custom layout, one can simply provide an ASCII string that describes the environment configuration, along with an optional list of possible recipes. If a list of recipes is not explicitly provided, a list of all possible combinations of ingredients available in the layout is used by default. The agent view radius can be configured independently of the layout; however, layouts may be designed with a particular radius in mind to create specific challenges.

```
1  # Define the layout string
2  overcookedv2_demo = """
3  WWPWW
4  0A A1
5  L   R
6  WBWXW
7  """
8  # Define possible recipes
9  recipes = [
10     [0,0,1],
11     [0,1,1]
12 ]
13 # Create the layout
14 layout = Layout.from_string(
15     overcookedv2_demo,
16     possible_recipes=recipes
17 )
```

| Char | Description |
|------|-------------|
| W | Wall (counter) |
| A | Agent |
| X | Delivery station |
| B | Plate (bowl) pile |
| P | Pot location |
| R | Recipe indicator |
| L | Button recipe indicator |
| 0-9 | Ingredient pile |
| (space) | Empty cell |

Figure 13: Example code block using the string to layout interface in the OvercookedV2 environment. This example defines the layout shown in Figure 1. The table on the right explains the symbols used in the layout grid.

# C   ADITIONAL MATERIAL FOR OVERCOOKED V2 EXPERIMENTS

## C.1   EXPERIMENTAL SETUP

In this section we outline the setting in which we conducted our experiments in OvercookedV2. We conducted all experiments in a partially observable setting with an agent view radius of two cells. Additionally, the starting positions of all agents were randomised at the beginning of each episode. The environment was configured to give negative rewards if the agents delivered the wrong dish. This setup prevents strategies in which the agent with access to all the necessary objects – ingredients, plates, pot, and delivery station – ignores coordination attempts from the other agent and alternately delivers the two possible recipes. With negative rewards, this strategy would yield an expected total return of zero.

### C.1.1   TRAINING

Initially, we attempted to train policies in the complex layouts of our new environment using architectures similar to those employed by Carroll et al. (2020). However, we found that these policies did not learn good policies in the new scenarios. Even in fully observable settings, agents struggled to perform optimally when conditioned on the recipe indicator. This led us to explore different architectures, drawing inspiration from ResNet (He et al., 2015) and SqueezeNet (Iandola et al., 2016). Ultimately, we discovered that incorporating three layers of 1x1 convolutions with 128, 128, and 8 channels respectively at the beginning of the network significantly improved performance. These 1x1 convolutions are highly efficient and effective since the observations in our environment are sparse. Adding these layers allows us to transform each cell of the observation grid and reduce its dimensionality before applying the more computationally expensive 3x3 convolutions. Following the 1x1 convolutions, we use three 2D convolutional layers with 16, 32, and 32 channels, each with a 3x3 kernel size, and apply zero-padding to the inputs. We use ReLU as the activation function after each convolutional layer. The output from the convolutional encoder is flattened and passed through a dense layer with 128 neurons, followed by LayerNorm (Ba et al., 2016).

Moreover, given that we operate in a partially observable environment, we added a memory component to our network architecture. Without this, agents would be unable to condition on their AOH, which is necessary, for example, to memorise the current recipe displayed on the recipe indicator block. To achieve this, we added a GRU (Chung et al., 2014) with a hidden size of 128, with the hidden state initialised to zeroes and reset at the termination of each episode. The output from the GRU is then passed through another dense layer with 128 neurons, which was used by the actor-network to parameterise a categorical distribution and by the critic to produce a value estimate via a projection layer. Experiments with more complex architectures did not yield substantial performance improvements and considerably increased training times. We trained our agents using PPO in the same setting as outlined in Appendix A.2.

### C.1.2   OTHER-PLAY

We implement other-play in Overcooked on top of our PPO implementation. Other-play relies on the knowledge of a class of symmetries $\Phi$ within the environment. In Overcooked, these symmetries can include action symmetries or ingredient permutations. For instance, in a layout with horizontal symmetry, the actions *up* and *down* could be considered symmetries. Similarly, ingredient permutations, or possibly permutations of subsets of the available ingredients, can form a symmetry class as defined by other-play. A limitation of other-play is that the set of symmetries must be known beforehand. This is particularly relevant in Overcooked, where we cannot define a universal set of symmetries for the entire environment; instead, symmetries depend on the specific layout.

In the layouts where we apply other-play in this section, we have two possible recipes, each consisting of three instances of the same ingredient type. We permute these two ingredients while keeping all other ingredients fixed. At the start of each episode, an ingredient permutation $\phi$ is randomly drawn from the set of symmetries $\Phi$ for each agent. This permutation is then applied to each agent's observation, permuting all observation encodings of ingredients, dishes, and ingredient dispensers. We implemented these (partial) ingredient permutations as part of the environment code and store the permutation mappings for each agent as part of the game state. When evaluating ZSC performance, all agents receive the non-permuted observations.

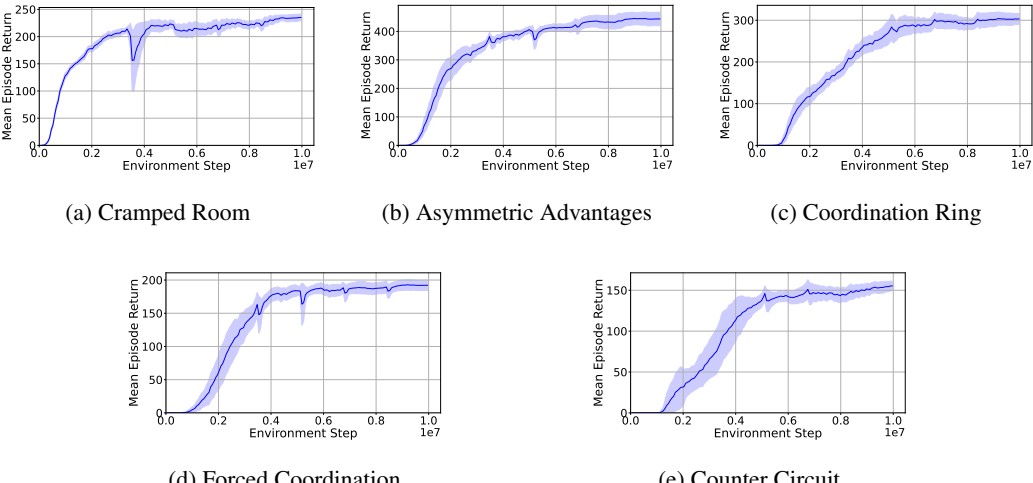

Figure 14: Training results on partially observable versions of the original Overcooked layouts with random starting positions in OvercookedV2. Plots show the mean and standard deviation across 10 seeds.

### C.1.3 FICTITIOUS CO-PLAY

Fictitious co-play is an algorithm that allows training robust agents capable of adapting to different skill levels. For each FCP agent we train a population of eight self-play agents and collect three checkpoints during training, with the final checkpoint representing the agents' final state. These checkpoints were then used to train a common best response. The parameters of the collected checkpoints are frozen, and only the FCP agent is trained. The batch of checkpoint partners used for training in each iteration was chosen at random. Additionally, the agent controlled by the FCP policy in the environment was selected randomly in each iteration. Therefore, the FCP policy needs to be robust enough to handle all agents' perspectives.

### C.2 ADAPTED LAYOUTS

**Original Layouts.** The evaluation of our model in a partially observable version of the layouts from the original Overcooked benchmark, illustrated in Figure 21, shows strong performance. The hyper-parameters for all layouts are available in Appendix D.2.1. Our agents trained in self-play achieve similar or even superior results compared to policies trained in the fully observable versions of the original environment (Carroll et al., 2020). In the Asymmetric Advantages and Coordination Rings layouts, all trained policies make use of both pots present in the layout to cook multiple dishes simultaneously; a behavior that previous policies struggled with. Even in the Counter Circuit layout, agents successfully deliver dishes, despite the layout being significantly larger than others. The increased size presents a greater challenge for agents operating in a partially observable environment. Furthermore, random position initialization can place agents far from the ingredients at the start, requiring them to be robust in navigating such situations. These layouts effectively demonstrate the capabilities of our policies in a partially observable setting; however, they do not yet present truly challenging coordination scenarios. Most of the layouts are relatively small, and except for the Asymmetric Advantages layout, agents are located in the same room, which diminishes the impact of partial observability. To create more interesting and challenging coordination scenarios, it is essential to introduce asymmetric information.

**Adapted Multi-Ingredient Layouts.** We extend two of these layouts by introducing an additional ingredient and a recipe indicator. This modification requires our policies to take into account the recipe of the current episode to effectively deliver dishes. Our results, illustrated in Figure 15, show that in both adapted layouts, agents successfully condition on the recipe indicator. However, certain recipe combinations yield slightly lower rewards than others, which may be due to slight differences in the distance between the respective ingredients and pots. In the Cramped Room V2

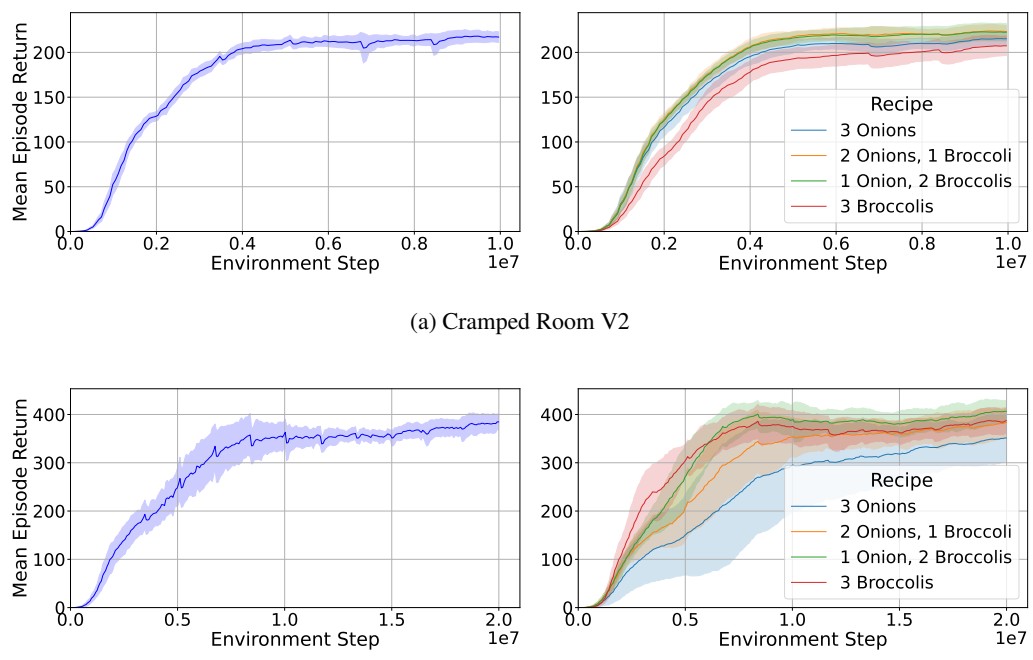

(a) Cramped Room V2

(b) Asymmetric Advantages V2 with recipe indicator on the right side

Figure 15: Training results on adapted Overcooked layouts in OvercookedV2. Both layouts feature two distinct ingredients (onions and broccoli) and a total of four possible recipes. The plots on the left display the mean total reward during training, while the plots on the right filter results to show only episodes where the respective recipe was selected. Each plot shows the mean and standard deviation across 10 seeds.

layout, both agents are located in the same room and can observe the recipe indicator if it is within their view radius. In the Asymmetric Advantages V2 layout, only the agent on the right can observe the indicator, requiring the other agent to wait for the recipe to be 'demoed' before delivering the correct dishes. These scenarios are more complex than the original benchmark layouts, but the coordination challenges remain limited, and there are straightforward solutions, such as 'demoing' the recipe, that work well in a ZSC setting. In the following sections, we will explore classes of layouts that present more complex coordination challenges with non-obvious solutions.

# D    HYPERPARAMETERS AND TRAINING CURVES

This appendix provides the hyperparameters used in our experiments.

## D.1    LIMITATIONS OF OVERCOOKED

### D.1.1    HYPERPARAMETERS

| Parameter | Value |
|---|---|
| ACTIVATION | relu |
| CLIP_EPS | 0.2 |
| CNN_FEATURES | 32 |
| ENT_COEF | 0.04 |
| FC_DIM_SIZE | 64 |
| GAE_LAMBDA | 0.95 |
| GAMMA | 0.99 |
| LR | 0.0004 |
| LR_WARMUP | 0.05 |
| MAX_GRAD_NORM | 0.5 |
| NUM_ENVS | 64 |
| NUM_MINIBATCHES | 16 |
| NUM_STEPS | 256 |
| REW_SHAPING_HORIZON | 5000000 |
| SCALE_CLIP_EPS | False |
| TOTAL_TIMESTEPS | 10000000 |
| UPDATE_EPOCHS | 4 |
| VF_COEF | 0.5 |

Table 3: Hyperparameters for the layouts: Cramped Room, Asymmetric advantages, Coordination Ring, Forced Coordination and Counter Circuit.

### D.1.2 TRAINING RESULTS

In this subsection we present training curves for both the standard and state-augmented settings across all layouts.

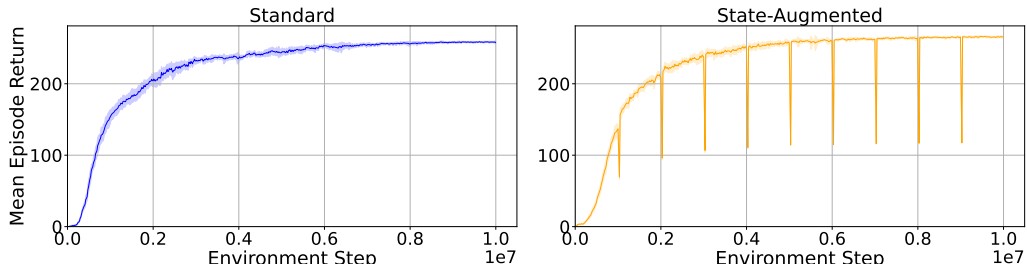

Figure 16: Training results for the limitations experiment in the Cramped Room layout. The plot on the left displays the mean total reward during training in the standard setting, while the plot on the right shows the state augmented setting. Each plot shows the mean and standard deviation across 10 seeds.

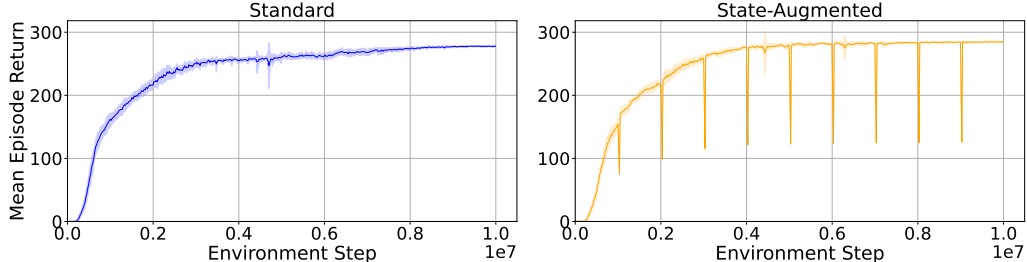

Figure 17: Training results for the limitations experiment in the Asymmetric Advanteges layout. The plot on the left displays the mean total reward during training in the standard setting, while the plot on the right shows the state augmented setting. Each plot shows the mean and standard deviation across 10 seeds.

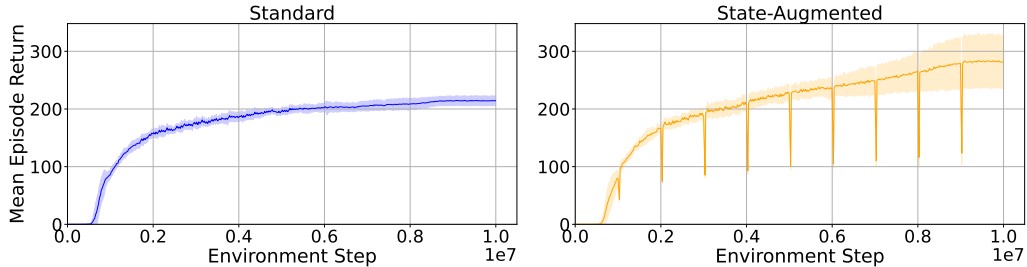

Figure 18: Training results for the limitations experiment in the Coordination Ring layout. The plot on the left displays the mean total reward during training in the standard setting, while the plot on the right shows the state augmented setting. Each plot shows the mean and standard deviation across 10 seeds.

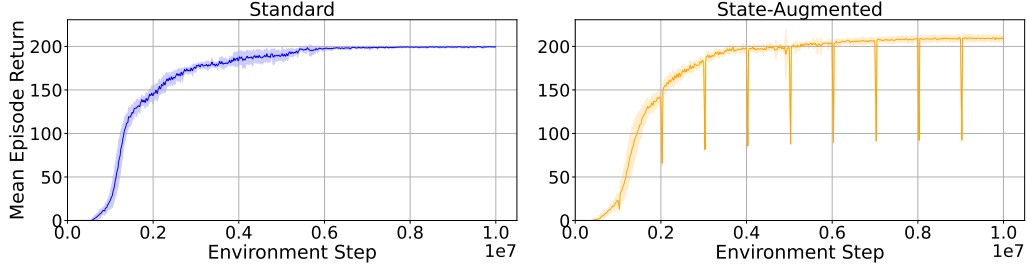

Figure 19: Training results for the limitations experiment in the Forced Coordination layout. The plot on the left displays the mean total reward during training in the standard setting, while the plot on the right shows the state augmented setting. Each plot shows the mean and standard deviation across 10 seeds.

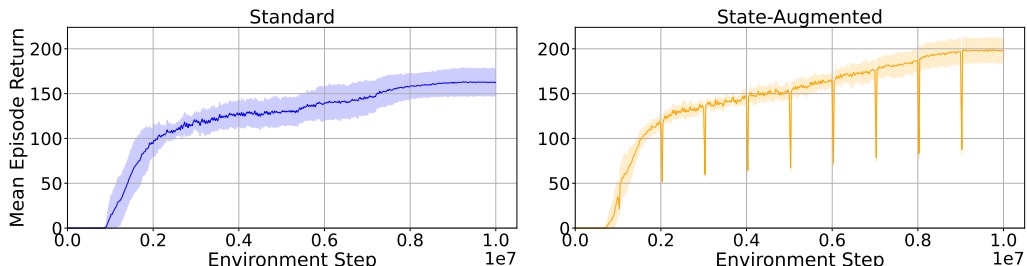

Figure 20: Training results for the limitations experiment in the Counter Circuit layout. The plot on the left displays the mean total reward during training in the standard setting, while the plot on the right shows the state augmented setting. Each plot shows the mean and standard deviation across 10 seeds.

## D.2 Experiments in OvercookedV2

### D.2.1 Adapted Layouts

| Parameter | Value |
|---|---|
| ACTIVATION | relu |
| ANNEAL_LR | True |
| CLIP_EPS | 0.2 |
| CNN_FEATURES | 32 |
| ENT_COEF | 0.01 |
| FC_DIM_SIZE | 128 |
| GAE_LAMBDA | 0.95 |
| GAMMA | 0.99 |
| GRU_HIDDEN_DIM | 128 |
| LR | 0.0005 |
| LR_WARMUP | 0.05 |
| MAX_GRAD_NORM | 0.25 |
| MINIBATCH_SIZE | 2048 |
| NUM_ENVS | 256 |
| NUM_MINIBATCHES | 64 |
| NUM_STEPS | 256 |
| REW_SHAPING_HORIZON | 5000000 |
| SCALE_CLIP_EPS | False |
| TOTAL_TIMESTEPS | 10000000 |
| UPDATE_EPOCHS | 4 |
| VF_COEF | 0.5 |

Table 4: Hyperparameters for layouts: Cramped Room, Asymmetric advantages, Coordination Ring, Forced Coordination and Counter Circuit.

| Parameter | Cramped Room V2 | Asymm. Advantages V2 |
|---|---|---|
| ACTIVATION | relu | relu |
| ANNEAL_LR | True | True |
| CLIP_EPS | 0.2 | 0.2 |
| CNN_FEATURES | 32 | 32 |
| ENT_COEF | 0.01 | 0.01 |
| FC_DIM_SIZE | 128 | 128 |
| GAE_LAMBDA | 0.95 | 0.95 |
| GAMMA | 0.99 | 0.99 |
| GRU_HIDDEN_DIM | 128 | 128 |
| LR | 0.0005 | 0.0005 |
| LR_WARMUP | 0.001 | 0.001 |
| MAX_GRAD_NORM | 0.25 | 0.25 |
| NUM_ENVS | 256 | 256 |
| NUM_MINIBATCHES | 64 | 64 |
| NUM_STEPS | 256 | 256 |
| REW_SHAPING_HORIZON | 5000000 | 10000000 |
| SCALE_CLIP_EPS | False | False |
| TOTAL_TIMESTEPS | 10000000 | 20000000 |
| UPDATE_EPOCHS | 4 | 4 |
| VF_COEF | 0.5 | 0.5 |

Table 5: Hyperparameters for the layouts Cramped Room V2 and Asymmetric Advantages V2.

## D.2.2 GROUNDED COORDINATION

| Environment Parameter | Value |
|---|---|
| agent_view_size | 2 |
| layout | grounded_coord_{simple/ring} |
| negative_rewards | True |
| random_agent_positions | True |
| sample_recipe_on_delivery | True |
| ENV_NAME | overcooked_v2 |

Table 6: Environment configuration for the Grounded Coordination layouts.

| Parameter | Self-Play | State-Augmented | Other-Play | Fictitious Co-Play |
|---|---|---|---|---|
| ACTIVATION | relu | relu | relu | relu |
| ANNEAL_LR | True | True | True | True |
| CLIP_EPS | 0.2 | 0.2 | 0.2 | 0.2 |
| CNN_FEATURES | 32 | 32 | 32 | 32 |
| ENT_COEF | 0.01 | 0.01 | 0.02 | 0.04 |
| FC_DIM_SIZE | 128 | 128 | 128 | 128 |
| GAE_LAMBDA | 0.95 | 0.95 | 0.95 | 0.9 |
| GAMMA | 0.99 | 0.99 | 0.99 | 0.99 |
| GRU_HIDDEN_DIM | 128 | 128 | 128 | 128 |
| LR | 0.00025 | 0.00025 | 0.00025 | 0.0007 |
| LR_WARMUP | 0.05 | 0.05 | 0.05 | 0.05 |
| MAX_GRAD_NORM | 0.25 | 0.25 | 0.25 | 0.25 |
| NUM_ENVS | 256 | 256 | 64 | 256 |
| NUM_MINIBATCHES | 64 | 64 | 64 | 64 |
| NUM_STEPS | 256 | 256 | 256 | 256 |
| REW_SHAPING_HORIZON | 15000000 | 15000000 | 25000000 | 15000000 |
| SCALE_CLIP_EPS | False | False | False | False |
| TOTAL_TIMESTEPS | 30000000 | 30000000 | 50000000 | 30000000 |
| UPDATE_EPOCHS | 4 | 4 | 4 | 4 |
| VF_COEF | 0.5 | 0.5 | 0.5 | 0.5 |

Table 7: Hyperparameters for Self-Play, State-Augmented, Other-Play and Fictitious Co-Play in the Grounded Coordination layouts layouts.

### D.2.3 TEST-TIME PROTOCOL FORMATION

| Environment Parameter | Value |
|---|---|
| agent_view_size | 2 |
| indicate_successful_delivery | True |
| layout | test_time_{simple/ring} |
| negative_rewards | True |
| random_agent_positions | True |
| sample_recipe_on_delivery | True |
| ENV_NAME | overcooked_v2 |

Table 8: Environment configuration for the Test Time Protocol Formation layouts.

| Parameter | Self-Play | State-Augmented | Other-Play | Fictitious Co-Play |
|---|---|---|---|---|
| ACTIVATION | relu | relu | relu | relu |
| ANNEAL_LR | True | True | True | True |
| CLIP_EPS | 0.2 | 0.2 | 0.2 | 0.2 |
| CNN_FEATURES | 32 | 32 | 32 | 32 |
| ENT_COEF | 0.01 | 0.01 | 0.02 | 0.04 |
| FC_DIM_SIZE | 128 | 128 | 128 | 128 |
| GAE_LAMBDA | 0.95 | 0.95 | 0.95 | 0.9 |
| GAMMA | 0.99 | 0.99 | 0.99 | 0.99 |
| GRU_HIDDEN_DIM | 128 | 128 | 128 | 128 |
| LR | 0.00025 | 0.00025 | 0.00025 | 0.0007 |
| LR_WARMUP | 0.05 | 0.05 | 0.05 | 0.05 |
| MAX_GRAD_NORM | 0.25 | 0.25 | 0.25 | 0.25 |
| NUM_ENVS | 256 | 256 | 64 | 256 |
| NUM_MINIBATCHES | 64 | 64 | 64 | 64 |
| NUM_STEPS | 256 | 256 | 256 | 256 |
| REW_SHAPING_HORIZON | 15000000 | 15000000 | 25000000 | 15000000 |
| SCALE_CLIP_EPS | False | False | False | False |
| TOTAL_TIMESTEPS | 30000000 | 30000000 | 50000000 | 30000000 |
| UPDATE_EPOCHS | 4 | 4 | 4 | 4 |
| VF_COEF | 0.5 | 0.5 | 0.5 | 0.5 |

Table 9: Hyperparameters for Self-Play, State-Augmented, Other-Play and Fictitious Co-Play in the Test Time Protocol layouts.

### D.2.4 DEMO COOK

| Environment Parameter | Value |
|---|---|
| agent_view_size | 2 |
| layout | test_time_{simple/wide} |
| negative_rewards | True |
| random_agent_positions | True |
| sample_recipe_on_delivery | True |
| ENV_NAME | overcooked_v2 |

Table 10: Environment configuration for the Demo Cook layouts.

| Parameter | Self-Play | State-Augmented | Other-Play | Fictitious Co-Play |
|---|---|---|---|---|
| ACTIVATION | relu | relu | relu | relu |
| ANNEAL_LR | True | True | True | True |
| CLIP_EPS | 0.2 | 0.2 | 0.2 | 0.2 |
| CNN_FEATURES | 32 | 32 | 32 | 32 |
| ENT_COEF | 0.01 | 0.01 | 0.02 | 0.04 |
| FC_DIM_SIZE | 128 | 128 | 128 | 128 |
| GAE_LAMBDA | 0.95 | 0.95 | 0.95 | 0.9 |
| GAMMA | 0.99 | 0.99 | 0.99 | 0.99 |
| GRU_HIDDEN_DIM | 128 | 128 | 128 | 128 |
| LR | 0.00025 | 0.00025 | 0.00025 | 0.0007 |
| LR_WARMUP | 0.05 | 0.05 | 0.05 | 0.05 |
| MAX_GRAD_NORM | 0.25 | 0.25 | 0.25 | 0.25 |
| NUM_ENVS | 256 | 256 | 64 | 256 |
| NUM_MINIBATCHES | 64 | 64 | 64 | 64 |
| NUM_STEPS | 256 | 256 | 256 | 256 |
| REW_SHAPING_HORIZON | 15000000 | 15000000 | 25000000 | 15000000 |
| SCALE_CLIP_EPS | False | False | False | False |
| TOTAL_TIMESTEPS | 30000000 | 30000000 | 50000000 | 30000000 |
| UPDATE_EPOCHS | 4 | 4 | 4 | 4 |
| VF_COEF | 0.5 | 0.5 | 0.5 | 0.5 |

Table 11: Hyperparameters for Self-Play, State-Augmented, Other-Play and Fictitious Co-Play in the Demo Cook layouts.

## D.3 FCP TRAINING CURVES

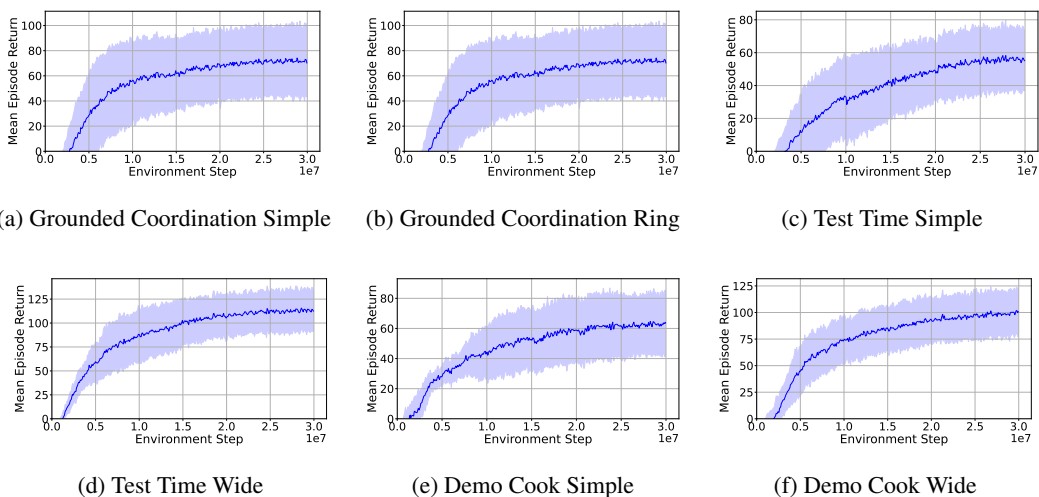

(a) Grounded Coordination Simple     (b) Grounded Coordination Ring     (c) Test Time Simple

(d) Test Time Wide         (e) Demo Cook Simple        (f) Demo Cook Wide

Figure 21: Training results for the FCP experiment in OvercookedV2 layouts. Each plot shows the mean and standard deviation across 10 seeds.

# E    FUTURE WORK

Future work should focus on evaluating additional ZSC methods in OvercookedV2. Our experiments demonstrated that state-of-the-art ZSC methods struggle with the complex coordination challenges present in OvercookedV2. While algorithms such as off-belief learning (Hu et al., 2021a) might offer improvements in some scenarios, fundamental challenges remain, particularly for test-time adaptation. Humans can adapt their strategies on the fly and coordinate with unknown partners from very few interactions, whereas RL is notoriously sample-inefficient (Hejna & Sadigh, 2022). Humans use their theory of mind to quickly reason about their partners and establish common conventions. Developing agents with this capability is required for effective human-AI coordination and should be a key focus for future ZSC research.

We plan to expand OvercookedV2 with new, challenging scenarios that test even more aspects of coordination. Specifically, we intend to design more layouts that pose complex coordination challenges. By introducing harder ZSC scenarios, we aim to reveal further gaps in existing approaches and drive additional research in this domain. Moreover, the grid-world nature of OvercookedV2 and its simple environment creation interface provide an ideal platform for exploring UED (Dennis et al., 2021). UED would enable the automatic generation of increasingly challenging coordination scenarios, creating a useful distribution of training scenarios. This also opens up new avenues for investigating the generalisation capabilities of ZSC methods, particularly in cooperating with novel partners in previously unseen layouts.

Lastly, while our primary focus in this work has been ZSC, our findings also apply to ad-hoc settings, where the original Overcooked benchmark has been commonly used. The generalisation challenge in ad-hoc settings is even greater due to the increased variability in skill levels and strategies compared to AI agents trained with the same method. For this reason, Overcooked was able to provide a challenging benchmark for ad-hoc coordination. However, the additional complexity that allows us to create scenarios requiring true coordination in OvercookedV2 will also be valuable for benchmarking the next generation of ad-hoc coordination methods. Furthermore, the partial observability in OvercookedV2 allows methods such as ADVERSITY to train a diverse population of "reasonable" agents which could be used for evaluation (Cui et al., 2023). We already provide an interactive interface for humans to interact with our new environment, and future work should focus on benchmarking human-AI coordination in OvercookedV2.

