# OpenReview forum: "OvercookedV2: Rethinking Overcooked for Zero-Shot Coordination"
_ICLR.cc/2025/Conference — ICLR 2025 Poster_

### Official Review · Reviewer_GLBf · 2024-10-28

**Soundness:** 3
**Presentation:** 4
**Contribution:** 3
**Rating:** 6
**Confidence:** 4

**Summary:**

This paper discusses limitations of the Overcooked benchmark, a widely adopted benchmark for zero shot coordination (ZSC), and proposes a new benchmark named OvercookedV2. The newly proposed benchmark better tests agents' coordinaiton skills and is less affected by the state coverage during agents' training (which significantly influences the test results on the Overcooked). In this way, OvercookedV2 mostly focuses on coordination between agents rather than generalization of the learned policy.

**Strengths:**

1. Zero shot coordination (ZSC) is an important topic

2. The Overcooked, the most popular benchmark for ZSC, is out-dated. It is even fair to say that challenges in this environment have already been solved. The proposed OvercookedV2 poses new challenges for studies on coordination (not only ZSC but also more general studies on agent coordination).

3. Thorough study on drawbacks of the Overcooked is given.

**Weaknesses:**

1. Insufficient evaluations of current ZSC methods on the new environment.

2. Some grammatical mistakes, e.g. 'where agents independently trained agents can adopt similar strategies but with arbitrary variations in execution'

**Questions:**

1. In the paper, you mentioned that in the Overcooked, ZSC failure can be attributed poor state coverage rather than more sophisticated coordination challenges. In this way, the Overcooked mainly evaluates policy generalization rather than coordination. So I wonder what is your definition of 'coordination'? Because I think generalizing to new partners is almost equal to ZSC.

2. Could you include more ZSC baselines in the paper. Other-play relies on pre-known symmetries in the environment so it may not work in the new environment. But there are also many ZSC methods that I believe are completely data-driven [1], [2], [3], [4].

[1] Strouse, D. J., et al. "Collaborating with humans without human data." Advances in Neural Information Processing Systems 34 (2021): 14502-14515.

[2] Zhao, Rui, et al. "Maximum entropy population-based training for zero-shot human-ai coordination." Proceedings of the AAAI Conference on Artificial Intelligence. Vol. 37. No. 5. 2023.

[3] Lou, Xingzhou, et al. "Pecan: Leveraging policy ensemble for context-aware zero-shot human-ai coordination." arXiv preprint arXiv:2301.06387 (2023).

[4] Yan, Xue, et al. "An efficient end-to-end training approach for zero-shot human-AI coordination." Advances in Neural Information Processing Systems 36 (2024).

3. I wonder whether you plan to integrate human-AI coordination into OvercookedV2, since another important advantage of the Overcooked is its support for human-AI interaction.

---

> ### Author Response · Authors · 2024-11-20
>
> We thank the time and effort reviewer GLBf has invested in reviewing our paper.
>
> ### Weaknesses
>
> > 1. Insufficient evaluations of current ZSC methods on the new environment.
>
> We are currently running additional experiments with the Fictitious Co-Play [1] algorithm and will update the paper accordingly. Furthermore, we will outline how OBL [4] could potentially solve certain layouts, however the algorithm implementation is highly complex and beyond the scope of this work.
>
>
> > 2. Some grammatical mistakes, e.g. 'where agents independently trained agents can adopt similar strategies but with arbitrary variations in execution'
>
> We addressed this particular grammatical mistake and are going over the paper to potentially polish the writing.
>
> ### Questions
> > 1. In the paper, you mentioned that in the Overcooked, ZSC failure can be attributed poor state coverage rather than more sophisticated coordination challenges. In this way, the Overcooked mainly evaluates policy generalization rather than coordination. So I wonder what is your definition of 'coordination'? Because I think generalizing to new partners is almost equal to ZSC.
>
> Here we refer to generalisation in the sense of state-coverage. Our results show that mere state-coverage is sufficient to virtually solve Overcooked, just like it is sufficient to solve the button game [2]. This is in opposition to other games where state coverage itself does not lead to strong coordination, like in Hanabi [3], the cat-dog problem [4], or other toy problems.
>
>
> > 2. Could you include more ZSC baselines in the paper. Other-play relies on pre-known symmetries in the environment so it may not work in the new environment. But there are also many ZSC methods that I believe are completely data-driven.
>
> As pointed out above, we are currently running additional experiments with the Fictitious Co-Play [1] algorithm and will update the paper accordingly. We agree that further baselines would be good and are working to implement additional ones.
>
>
> > 3. I wonder whether you plan to integrate human-AI coordination into OvercookedV2, since another important advantage of the Overcooked is its support for human-AI interaction.
>
> We have a human-AI interface available, which is described in the Appendix. We will add more details to the description as well as  screenshots of what it looks like.
> We hope to conduct human-AI studies in the future,  but this is out of scope now.
>
>
>
> [1] Strouse, D. J., et al. "Collaborating with humans without human data." Advances in Neural Information Processing Systems 34 (2021): 14502-14515.
>
> [2] Section 4, OvercookedV2: Rethinking Overcooked for ZSC
>
> [3] Bard, Nolan, et al. "The hanabi challenge: A new frontier for ai research." Artificial Intelligence 280 (2020): 103216.
>
> [4] Hu, Hengyuan, et al. "Off-belief learning." International Conference on Machine Learning. PMLR, 2021.

---

> > ### Comment · Area_Chair_8hyz · 2024-11-24
> > **Please respond to rebuttal ASAP**
> >
> > Dear reviewer,
> > The process only works if we engage in discussion. Can you please respond to the rebuttal provided by the authors ASAP?

---

> > > ### Comment · Reviewer_GLBf · 2024-11-25
> > >
> > > Thanks for the response provided by the authors. The response repeats that generalization to new partners is not coordination but more like state-coverage, but I am still confused about what the authors' opinion on the definition of coordination and why Overcooked-V2 evaluates coordinates but not state coverage. Are the states hard to cover?

---

> > > > ### Author Response · Authors · 2024-11-26
> > > >
> > > > Thank you for your comment! In OvercookedV1, we show that state-coverage alone is sufficient for agents to perform well in a ZSC setting. While state-coverage is one aspect of coordination, it is only one of the challenges involved in collaborating with unknown partners. As such, OvercookedV1 is not a suitable benchmark for evaluating ZSC.
> > > > OvercookedV2 provides richer coordination challenges. While state-coverage *remains* part of the challenge, it is no longer sufficient to solve the benchmark. OvercookedV2 enables us to design scenarios that cannot be solved by state-coverage alone. For example, we can create scenarios in which agents need to rely on grounded communication and/or test-time protocol formation. This makes OvercookedV2 a much more comprehensive benchmark for evaluating ZSC, allowing us to create more complex and realistic scenarios.
> > > > We will make this clearer in the paper.

---

> > > > > ### Comment · Reviewer_GLBf · 2024-11-27
> > > > >
> > > > > Thanks for your clarification. Could you please provide the results of FCP on Overcooked V2 so that I can judge whether methods that worked on Overcooked V1 cannot solve the new challenges proposed in Overcooked V2

---

> > > > > > ### Author Response · Authors · 2024-11-29
> > > > > >
> > > > > > We are still conducting FCP experiments for OvercookedV2. These experiments involve training a large population of agents, which requires substantial computational time, especially as we need to repeat this process for 10 independent seeds for a ZSC evaluation.
> > > > > > Nevertheless, we are already observing interesting results. For example, in the Grounded Coordination Simple layout, a trained FCP agent performs significantly worse (score of ~40) than any individual agent in the population (scores >100). The FCP agent must adapt to various partners within the population, which is challenging due to the partial observability of the environment and the requirement for communication.
> > > > > > In an additional experiment with a smaller population, where agents are selected to be inter-compatible, we observe significantly higher FCP scores of ~100. This highlights how FCP struggles with complex coordination scenarios like those in OvercookedV2.
> > > > > > We will include a detailed analysis of FCP performance in OvercookedV2 in the paper.

---

### Official Review · Reviewer_Fs42 · 2024-11-01

**Soundness:** 3
**Presentation:** 3
**Contribution:** 3
**Rating:** 5
**Confidence:** 5

**Summary:**

The study of Zero-shot Human-AI coordination is important within the MARL community. Prior work has focused on the benchmark Overcooked, and shown how naive methods such as self-play fail because they form arbitrary conventions that don't generalize. However, this work shows that by merely increasing the state coverage self-play agents are exposed to, they can close the SP-XP gap. As such, the authors conclude the original 5 layouts are not sufficient coordination challenges. They then introduce a new benchmark which extends Overcooked by introducing asymmetric information and stochasticity, and show that state coverage is insufficient to coordinate on this new setting.

**Strengths:**

This is a compelling idea and it was very nice to see how for self-play agents, state coverage on the original 5 layouts essentially solved the generalization gap. The authors did a nice job explaining how their new environment introduces novel challenges to the Overcooked literature through communication and partial observability, as well as limitations of self-play on these new layouts.

**Weaknesses:**

It would've been nice for all evaluations of the state-augmented agents on the default layouts to see how well they play with humans. If state coverage is really the key to removing the SP-XP gap, then we should see performance with humans be higher than conventional SP method playing with humans. Otherwise, it is not clear that state-coverage alone is getting at the ZSC capabilities methods like population-based training tries to learn. A similarly insightful analysis would be comparing how well the state-augmented agents on the default layouts play with other algorithms (SP, MEP, E3T, etc.) that are SOTA for ZSC on overcooked. If state-augmented agents can perform at SOTA levels or close to that, it would be more compelling evidence that the issue with generalization on the original problems was just state coverage.

For the new environments, it would be compelling to see baselines such as MEP or E3T, which are some of the most successful algorithms for ZSC in Overcooked (the former being population based, the latter being self-play based). MEP and other population-based methods in particular try to simulate strategy diversity (by generating diversity in trajectories of states visited), so if these methods fail on your new benchmark it would be especially compelling as a new challenge for the MARL community to try to tackle.

**Questions:**

The "environment creation via ASCII text" was listed as a contribution. How does this differ from the same functionality the original JaxMARL codebase already included beyond the additional object predicates from your new layout?

Similarly, you listed there was an interactive interface for humans to play on the new environment. JaxMARL included this for Overcooked as well, so how is your interface different? If it's easy to benchmark human-AI coordination would you be able to show preliminary results on the new benchmark?

---

> ### Author Response · Authors · 2024-11-20
>
> We thank the time and effort reviewer Fs42 has invested in reviewing our paper.
>
>
> ### Weaknesses
>
> > It would've been nice for all evaluations of the state-augmented agents on the default layouts to see how well they play with humans. If state coverage is really the key to removing the SP-XP gap, then we should see performance with humans be higher than conventional SP method playing with humans. Otherwise, it is not clear that state-coverage alone is getting at the ZSC capabilities methods like population-based training tries to learn.
>
> We remind that ZSC is a formal setting introduced in Hu et al. that requires constructing an algorithm that allows agents to be trained independently yet coordinate effectively at test time. It is a prerequisite to human-AI coordination since it is the lowest bar to cross, but not necessarily sufficient.
> Regardless of whether SA agents can play with humans or not in Overcooked, that environment does not assess human-AI coordination or ZSC in stochastic partially observable settings.
> OvercookedV2 provides the ability to evaluate that, and we show a gap in ZSC, which necessarily implies a gap in human-AI coordination as well.
>
>
> > A similarly insightful analysis would be comparing how well the state-augmented agents on the default layouts play with other algorithms (SP, MEP, E3T, etc.) that are SOTA for ZSC on overcooked. If state-augmented agents can perform at SOTA levels or close to that, it would be more compelling evidence that the issue with generalization on the original problems was just state coverage.
>
> The aforementioned algorithms may themselves not be robust to unseen states. Low XP scores might therefore come from those other algorithms, rather than from the SA agents. For an extreme example of this see ADVERSITY [2].
>
> > For the new environments, it would be compelling to see baselines such as MEP or E3T, which are some of the most successful algorithms for ZSC in Overcooked (the former being population based, the latter being self-play based). MEP and other population-based methods in particular try to simulate strategy diversity (by generating diversity in trajectories of states visited), so if these methods fail on your new benchmark it would be especially compelling as a new challenge for the MARL community to try to tackle.
>
> We agree that additional baselines are good and we are currently running additional experiments with the Fictitious Co-Play [3] algorithm and will update the paper accordingly. Furthermore, we will outline how OBL [4] could potentially solve certain layouts, however the algorithm implementation is highly complex and beyond the scope of this work.
>
>
> ### Questions
>
> > The "environment creation via ASCII text" was listed as a contribution. How does this differ from the same functionality the original JaxMARL codebase already included beyond the additional object predicates from your new layout?
>
> The interface is similar to the one in the original JaxMARL codebase, however it is modified to allow for the additional objects introduced in OvercookedV2. Moreover, the interface allows additional parameters such as possible recipes.
>
>
> > Similarly, you listed there was an interactive interface for humans to play on the new environment. JaxMARL included this for Overcooked as well, so how is your interface different? If it's easy to benchmark human-AI coordination would you be able to show preliminary results on the new benchmark?
>
> We modified the interface to allow players to collaborate with trained policies, this was not possible previously. Moreover, the interface is extended to include all new items introduced in OvercookedV2. If partial observability is enabled, it is possible to also restrict the view radius of the player to simulate what the agent would see. This enables future human-AI experiments, however this is beyond the scope of this paper.
>
>
>
> [1] Hu, Hengyuan, et al. "“other-play” for zero-shot coordination." International Conference on Machine Learning. PMLR, 2020.
> [2] Cui, Brandon, et al. "Adversarial diversity in hanabi." The Eleventh International Conference on Learning Representations. 2023.
> [3] Strouse, D. J., et al. "Collaborating with humans without human data." Advances in Neural Information Processing Systems 34 (2021): 14502-14515.
> [4] Hu, Hengyuan, et al. "Off-belief learning." International Conference on Machine Learning. PMLR, 2021.

---

> > ### Comment · Reviewer_Fs42 · 2024-11-21
> >
> > '''
> > The aforementioned algorithms may themselves not be robust to unseen states. Low XP scores might therefore come from those other algorithms, rather than from the SA agents. For an extreme example of this see ADVERSITY [2].
> > '''  - This is fair if you only compare 2 algorithms (i.e. SA with naive Self-play). However, if you make an n by n matrix, where 1...n is a different algorithm, and show cross play performance in each of these, the best algorithm should on average be better when playing with other algorithms. For instance, have each of (self-play, FCP, and SA) play with variants of themselves. If SA playing with other algorithms does better than Self-play playing with other algorithms, then this is a good indicator it's better at ZSC with agents learning under different algorithms. If you don't at least have some analysis showing how SA plays with other algorithms or people, it's hard to say that the generalization gap is solved and the original Overcooked environment is not challenging, since it is possible all SA seeds could have converged upon a similar norm.
> >
> >    - Moreover, it's not clear that human evaluations are beyond the scope of this paper. If you are introducing a new challenge, shouldn't there be a baseline for the type of performance we'd like our AI agents to achieve?
> >    - Thank you for clarifying how your human interface and layout generator builds on JaxMARL rather than completely writes a new one. In that case, I feel it would be best to edit the sections listing that as a contribution to be credit that prior work more and clearly lay out where the novelty is.

---

> > > ### Author Response · Authors · 2024-11-26
> > >
> > > We thank reviewer Fs42 for their comments.
> > >
> > > > If you don't at least have some analysis showing how SA plays with other algorithms or people, it's hard to say that the generalization gap is solved and the original Overcooked environment is not challenging, since it is possible all SA seeds could have converged upon a similar norm.
> > >
> > > If all seeds converge to a similar norm, this indicates that the range of viable strategies for achieving high returns is very limited. For comparison, consider the challenging coordination benchmark Hanabi, where ZSC evaluations with different seeds have struggled significantly [1, 2, 3]. In Hanabi, agents can adopt a wide variety of arbitrary conventions to achieve high returns. Our results suggest that in Overcooked, the diversity of fundamentally different high return strategies is much more limited. While this might still pose a challenge in human-AI scenarios, where agents must adapt to weaker partners, it does not provide a significant challenge for ZSC.
> > >
> > >
> > > > Moreover, it's not clear that human evaluations are beyond the scope of this paper. If you are introducing a new challenge, shouldn't there be a baseline for the type of performance we'd like our AI agents to achieve?
> > >
> > > In this paper we focus on the ZSC challenge. ZSC is a prerequisite to human-AI coordination since it is the lowest bar to cross, but not necessarily sufficient. We showed that the original Overcooked benchmark for ZSC can mostly be solved by state-coverage alone. While state-coverage is one aspect of coordination, it is only one of the challenges involved in collaborating with unknown partners. As such, OvercookedV1 is not a suitable benchmark for evaluating ZSC. Therefore we introduce OvercookedV2 which provides a richer coordination environment, in which we can evaluate ZSC algorithms. These findings and the advantages of OvercookedV2 can be transferred to human-AI coordination; however, in this paper we focus on ZSC.
> > >
> > > >Thank you for clarifying how your human interface and layout generator builds on JaxMARL rather than completely writes a new one. In that case, I feel it would be best to edit the sections listing that as a contribution to be credit that prior work more and clearly lay out where the novelty is.
> > >
> > > We will clarify this in the paper.
> > >
> > >
> > > [1] Hu, Hengyuan, et al. "Off-belief learning." International Conference on Machine Learning. PMLR, 2021.
> > > [2] Hu, Hengyuan, et al. "“other-play” for zero-shot coordination." International Conference on Machine Learning. PMLR, 2020.
> > > [3] Lupu, Andrei, et al. "Trajectory diversity for zero-shot coordination." International conference on machine learning. PMLR, 2021.

---

### Official Review · Reviewer_XSHM · 2024-11-02

**Soundness:** 3
**Presentation:** 1
**Contribution:** 2
**Rating:** 5
**Confidence:** 4

**Summary:**

A key challenge in zero-shot coordination (ZSC) is how to communicate with an agent that has never been seen before. Typical algorithms such as self-play settle on arbitrary communication protocols, which no longer work when playing with a never-before-seen agent that doesn’t have the same communication protocol. Instead, agents would ideally learn grounded communication protocols.

So far, ZSC algorithms have been tested primarily in Hanabi and Overcooked. However, in Overcooked, there is no need for grounded communication – indeed, many failures in Overcooked can be eliminated by simply increasing the diversity of states encountered during training – so it does not serve as a good test of the grounded communication aspect of ZSC.

To address this, the authors extend Overcooked to add features that induce partial observability, stochasticity, and asymmetric information across agents, and use these features to design six new handcrafted coordination challenges, which require grounded communication to perform well in the ZSC setting. Experiments demonstrate that the environment is challenging for existing algorithms, making it a good benchmark to hill climb on.

**Strengths:**

* As the paper notes, grounded communication for ZSC is an interesting area of research, and there is only one benchmark for it currently (Hanabi). Evaluating solely on Hanabi runs the risk of overfitting to the environment.
* The authors don’t note this, but I especially like that OvercookedV2 requires that grounded communication be done alongside other skills such as low-level motion planning (unlike Hanabi which is almost purely a communication challenge).
* I _think_ at least the simple versions of the handcrafted coordination challenges make sense as tests of different types of grounded communication, though I found it hard to evaluate due to missing details in the paper.

**Weaknesses:**

## State robustness is a part of ZSC

The authors argue that OvercookedV1 is a poor fit for testing ZSC because a lot of the failures are due to poor state robustness. However, I would argue that state robustness *is* a key challenge in ZSC. When playing a collaborative game with a novel partner, the partner’s actions may move into novel, off-distribution states; to be good at ZSC an agent must be robust to this. Algorithms like self-play naturally fail to produce state robustness because they exploit a single collaborative strategy that only visits a narrow part of state space; this is why algorithms like [Fictitious Co-Play](https://arxiv.org/abs/2110.08176) improve results – by increasing the diversity of partners, we increase the diversity of states that the agent is trained on, making it better at ZSC.

Based on Section 4, it seems that the authors are interested only in “fundamentally incompatible conventions”, giving this the name a “true coordination challenge”. The intended solution seems to be to base conventions off of actions that are meaningfully grounded in the environment dynamics, which I would call “grounded communication”, following [Hu et al](https://proceedings.mlr.press/v139/hu21c.html). This is a fine focus, but it should not be confused with ZSC – it is a particular subskill that is required to do well at ZSC.

Relatedly, lines 141-152 discuss how “Partial observability and stochasticity are required for a problem to be truly multi-agent” because “there is an optimal joint policy where the only information required is the timestep and agent ID, which essentially boils down to a pre-determined sequence of actions”. While this is all technically accurate, this is irrelevant to the ZSC problem, where the whole point is that the agents do not get to coordinate ahead of time on an optimal joint policy. In fact, the [Overcooked video game](https://store.steampowered.com/app/448510/Overcooked/) that the benchmark is based on is itself (usually) fully observable, but nonetheless poses significant ZSC challenges, as anyone who has played it can attest.

I would recommend that the authors rewrite the paper to focus on the grounded communication problem, rather than ZSC more generally. The contributions of the paper make much more sense interpreted this way – indeed increased state robustness should not make much of a difference to grounded communication. Elsewhere in this review I have treated the paper as though it were discussing the problem of grounded communication rather than ZSC more broadly.

## Missing details

One of the key contributions of this work is the six new handcrafted coordination challenges, illustrated in Figure 5. However the paper provides fairly limited details about these layouts (including in the appendices, though perhaps I missed the details), making it very difficult to understand what the challenge in these environments is. I think I’ve inferred it based on the prose, but the paper should specify it directly (see also the subheading “Questions about the handcrafted layouts” under “Questions”). I would recommend that the authors put much more detail into Section 6.3, particularly discussing the intended solution to each of the layouts.

## Why is OvercookedV2 difficult?

The authors argue that OvercookedV2 is a good benchmark to test grounded communication. In support of this, they show that state augmentation alone is insufficient to close the gap between self-play and cross-play in OvercookedV2.

Firstly, this isn’t quite as obvious as the paper suggests. According to Table 1, for original Overcooked on Cramped Room and Asymmetric Advantages, there is approximately no gap to begin with, on Coordination Ring and Counter Circuit, the gap is approximately halved, and on Forced Coordination the gap is mostly eliminated. So statements like “By employing a state augmentation algorithm during training, we nearly close the SP-XP gap across all layouts” (line 67) are overclaims. In OvercookedV2, state augmentation doesn’t change the gap on three environments, but does substantially reduce the gap on the other three environments. So while state augmentation is certainly less effective for OvercookedV2, it isn’t ineffective.

However, the more important objection is that this doesn’t convince me that OvercookedV2 is limited by the ability to do grounded communication. OvercookedV2 adds significantly more complexity to the environment, resulting in a greatly expanded effective state space to be robust to. In particular, with partial observability, optimal policies now depend on _belief states_, which is an exponentially larger space than regular state space. It is plausible that state augmentation was enough to address state robustness in the relatively limited OvercookedV1 environments, but could not address state robustness in OvercookedV2.

To address this, the authors could conduct a qualitative analysis of the failures exhibited during cross-play in OvercookedV2. Do we see failures of grounded communication, or failures of robustness? Ideally the authors would extend the existing Overcooked demo linked in footnote 2 with their environments and policies, so readers could observe these failures themselves.

The authors could also investigate how the effectiveness of the state augmentation method changes with complexity of the environment, to predict how much the OvercookedV2 environments “should” be helped by state augmentation if they didn’t have grounded communication challenges, and compare that with actual performance. However, this is probably more effort than it is worth.

## Baselines

I believe that the current state of the art algorithm for grounded communication is Off Belief Learning (OBL), from which the authors get their Button game. The experimental results on OvercookedV2 should include OBL as well – it seems quite plausible that OBL succeeds where State Augmentation and Other-Play did not, particularly since Overcooked is a poor fit for the “symmetry-breaking” approach employed by Other-Play. (Though if the primary challenge for OvercookedV2 is still state robustness, as would be my guess, then OBL probably wouldn’t make much of a difference.)

Another relevant algorithm is [Fictitious Co-Play](https://arxiv.org/abs/2110.08176), though unlike OBL this is not targeted at grounded communication, and so is reasonable to exclude.

**Questions:**

## Questions on the general message

* Is it accurate to say that the focus of the paper is on grounded communication, rather than coordination more generally? (If not, why not?)
* Can you say more about the intended solutions for each of the handcrafted challenges?
* Do you think humans would solve these environments as intended, if they were given only the rules of the game and some time to familiarize themselves with the environment? (It would be interesting to do a human study to check.)

## Questions about the handcrafted layouts

Here are my current guesses about how the layouts work – are any of these incorrect?
* The white circle with red background is a recipe indicator block, the red circle is a button recipe indicator, and the green square is a delivery location. (Please add a legend to Figure 5 for future readers.)
* Agents are only provided a signal for successful delivery in the Test-Time Protocol Formation setups.
* The yellow (onion) and green (broccoli) ingredients are to be actually used in recipes, while the blue ingredient is available to enable arbitrary communication strategies.

What is the view radius for each of the environments? This matters to interpret the environments – for example, a view radius of 1 in Demo Cook would imply that the agents must communicate via counter placements or pot interactions, whereas a view radius of 2 or more would allow the agents to communicate via their movements as well.

(Incidentally, I would recommend the authors also consider communication through movement as a form of grounded communication – this has been studied under the term “legibility”, see e.g. [Dragan et al, 2013](https://www.ri.cmu.edu/pub_files/2013/3/legiilitypredictabilityIEEE.pdf).)

(Ideally the paper would also specify other hyperparameters such as the sizes of various positive and negative rewards, but they are not as crucial.)

## Other notes

* Please add some more clarification about the Button game somewhere (appendix is fine). I had to read the Colab to understand how it worked. (In particular, it is quite unintuitive that when Alice presses a button, it does not turn on a fixed light bulb, but rather which light bulb it turns on depends on whether the pet is a cat or a dog. Consider reskinning the environment to be more intuitive, e.g. maybe when Alice presses a button, it removes one of N screens on Bob’s side of the room, and behind the screen is a picture of the pet.)

## Overall take

I’ve recommended a Reject (3), but I want to note that I do like a lot of the work and think that it is reasonably close to a point where I’d want to accept it. Mostly what is needed is a significant rewrite, focusing more narrowly on grounded communication in particular, with less emphasis on state augmentation, and significantly more exposition of the handcrafted challenges. It will likely also require collecting some more evidence about the utility of the environment for grounded communication (most notably a qualitative analysis of failure modes, and having OBL as a baseline). This evidence may reveal issues with the handcrafted layouts, but if so I expect the authors will be able to design new layouts that fix the issues.

---

> ### Author Response · Authors · 2024-11-20
> **Rebuttal**
>
> We thank the reviewer XSHM for the time and effort invested in extensively reviewing our paper.
>
> ### State robustness is a part of ZSC
>
> > The intended solution seems to be to base conventions off of actions that are meaningfully grounded in the environment dynamics, which I would call “grounded communication,” following Hu et al.
>
> This is not what "grounded" means. Grounded communication is communication that contains verifiable information, independent of the speaker's intentions (i.e., grounded actions carry meaning even if taken by a purely random agent). In OvercookedV2, the recipe indicator acts as grounded communication since it communicates the correct recipe as soon as triggered, even if triggered randomly. However, OvercookedV2 purposely features layouts where grounded communication is impossible, such as the layouts *Test Time Simple/Wide* and *Demo Cook Simple/Wide*, where agents can only form protocols by moving in specific ways or placing items on a counter, but where those actions do not carry any information a priori.
> A way to solve these scenarios is to dynamically form an arbitrary protocol at test time through repetition. As such, OvercookedV2 is about more than just grounded communication.
>
> > While this is all technically accurate, this is irrelevant to the ZSC problem, where the whole point is that the agents do not get to coordinate ahead of time on an optimal joint policy. In fact, the Overcooked video game that the benchmark is based on is itself (usually) fully observable, but nonetheless poses significant ZSC challenges, as anyone who has played it can attest.
>
> The bigger point is that, in a fully observable setting, other agents can simply be reactive by observing what the other is doing and choosing the immediate best response. However, the scenario is challenging for humans because they have limited attention; so, although it is fully observable, players cannot attend to all information at all times. Furthermore, the video game introduces stochasticity by randomising recipes (unlike OvercookedV1).
>
> ### Missing Details
>
> We agree that a discussion of the intended solution for the handcrafted challenges is needed, and we will include those. Moreover, we will outline how OBL could help in each of the scenarios.
>
> ### Why is OvercookedV2 difficult?
>
> > In OvercookedV2, state augmentation doesn’t change the gap on three environments but does substantially reduce the gap on the other three environments. So while state augmentation is certainly less effective for OvercookedV2, it isn’t ineffective.
>
> It is correct that state augmentation still reduces the gap significantly in OvercookedV2, but not nearly as much as seen in OvercookedV1. OvercookedV2 can still be used to create simple environments, and indeed, *Demo Cook Simple* can be considered simpler than the others. However, it also allows for constructing much more complex and rich layouts.
>
> > However, the more important objection is that this doesn’t convince me that OvercookedV2 is limited by the ability to do grounded communication.
>
> Again, not all layouts we propose even contain the option for grounded communication. In layouts such as *Test Time Simple/Wide*, agents must form a protocol at test time through pattern-matching, as none of their actions carry verifiable information.
>
> > It is plausible that state augmentation was enough to address state robustness in the relatively limited OvercookedV1 environments but could not address state robustness in OvercookedV2.
>
> Yes, this is precisely correct. OvercookedV1 mostly depended on the state space (which is independent of the history or partner policy). In contrast, optimal policies in OvercookedV2 depend on belief states, which are determined by the history and partner policy. Therefore, belief state augmentation requires reasoning about potential partners, which is key to addressing coordination challenges.
>
> > To address this, the authors could conduct a qualitative analysis of the failures exhibited during cross-play in OvercookedV2.
>
> We agree that a qualitative analysis would be valuable, and we are including a discussion as well as making GIFs of these scenarios available.
>
> > The authors could also investigate how the effectiveness of the state augmentation method changes with the complexity of the environment.
>
> This raises the question: how do we measure complexity? We do have SA results for environments without grounded communication, such as *Test Time Simple/Wide*.
>
> ### Baselines
>
> We agree that additional baselines would be valuable and are currently running experiments with Fictitious Co-Play. Implementing OBL is a complex effort beyond the scope of this paper. However, we will outline how we expect OBL to perform in the handcrafted coordination scenarios.

---

> ### Author Response · Authors · 2024-11-20
> **Open Questions**
>
> ### Questions on the General Message
>
> > Is it accurate to say that the focus of the paper is on grounded communication, rather than coordination more generally? (If not, why not?)
>
> No, this is not accurate, as outlined above (see comments under "State robustness is a part of ZSC").
>
> > Can you say more about the intended solutions for each of the handcrafted challenges?
>
> Yes, we agree that there should be a more detailed explanation for each challenge. We will describe the intended solution and also outline how algorithms such as Off-Belief Learning (OBL) could solve certain scenarios.
>
> > Do you think humans would solve these environments as intended, if they were given only the rules of the game and some time to familiarize themselves with the environment? (It would be interesting to do a human study to check.)
>
> In the grounded coordination layouts, humans would most likely be able to make meaningful progress. For example, it is intuitive for humans to accept a negative reward of -5 to reveal the recipe. However, using this grounded information to perform even more optimally—by forming a protocol at test time—is significantly more challenging.
> Similarly, in the test-time layouts, there is no straightforward way to leverage grounded communication. Here, players would need to interact repeatedly to form a protocol, which humans should be capable of.
> Lastly, the demo cook layouts provide solutions that are intuitive for humans. Humans can reason about the meaning of others' actions and generally assume that the other actor is rational, which would lead them to successfully solve these layouts.
> However, conducting human studies is beyond the scope of this work. We hope that future work will explore human-AI coordination in OvercookedV2.
>
>
> ### Questions About the Handcrafted Layouts
>
> > Here are my current guesses about how the layouts work – are any of these incorrect?
>
> These are all correct. We will add clarification to Figure 5.
>
> > What is the view radius for each of the environments? This matters to interpret the environments – for example, a view radius of 1 in *Demo Cook* would imply that the agents must communicate via counter placements or pot interactions, whereas a view radius of 2 or more would allow the agents to communicate via their movements as well.
>
> The view radius for all handcrafted coordination challenges in the paper is 2. This is also stated in Appendix C.1. Communication through movement is possible in some layouts, such as the *Grounded Coordination Ring*.
>
> > (Incidentally, I would recommend the authors also consider communication through movement as a form of grounded communication – this has been studied under the term “legibility”; see, e.g., Dragan et al., 2013.)
>
> While movement can carry information (e.g., by moving toward a goal to indicate the intention of reaching it), it only conveys information if we assume the moving agent is approximately rational. A random agent performing the same movement would not communicate any bits of information.
>
> Note that our *Demo Cook* layouts specifically serve to study the emergence of meaning through actions directed at specific goals, such as depositing a onion item in the pot to communicate that the three onion recipe is the correct one.
>
> > (Ideally the paper would also specify other hyperparameters such as the sizes of various positive and negative rewards, but they are not as crucial.)
>
> The sizes of positive and negative rewards (+/- 20) are stated in Appendix B.2.
>
>
> ### Other Notes
>
> > Please add some more clarification about the Button game somewhere (appendix is fine).
>
> We will rewrite the explanation and include more details to make this clearer.

---

> > ### Comment · Area_Chair_8hyz · 2024-11-24
> > **Please respond to rebuttal asap**
> >
> > Dear reviewer,
> > The process only works if we engage in discussion. Can you please respond to the rebuttal provided by the authors ASAP?

---

> ### Comment · Reviewer_XSHM · 2024-11-24
> **Still confused at what the authors mean by ZSC**
>
> Thanks for agreeing to add baselines and more details about the layouts. I've increased my score to a 5.
>
> I still do not understand what your definition of zero-shot coordination is, if you think that "mere state robustness" does not count as a solution to ZSC. And it's not just me, Reviewer GLBf also has this confusion, saying _"So I wonder what is your definition of 'coordination'? Because I think generalizing to new partners is almost equal to ZSC."_
>
> One argument I am more sympathetic to is: the original Overcooked benchmark is almost saturated using a simple state robustness method. So, it is time to develop a harder benchmark. In this story, the state robustness algorithm is simply one algorithm for achieving ZSC, that works in the original Overcooked setting, but doesn't work for Overcookedv2.
>
> > The bigger point is that, in a fully observable setting, other agents can simply be reactive by observing what the other is doing and choosing the immediate best response.
>
> I'm not sure exactly what you are trying to say here, but I think it is probably untrue. There is no unique "best response" that is independent of the other agent, even in a fully observable setting. This was the main point in the original Overcooked paper (Carroll et al) -- Figure 1 demonstrates how incorrect beliefs about the partner agent's future actions can lead to bad outcomes.
>
> > While movement can carry information (e.g., by moving toward a goal to indicate the intention of reaching it), it only conveys information if we assume the moving agent is approximately rational. A random agent performing the same movement would not communicate any bits of information.
>
> Yes, of course this depends on some minimal rationality assumptions, but this is also true of test-time protocol formation more generally. A random agent doesn't communicate information through its movement -- but you also can't form any test-time protocol with a random agent.

---

> ### Author Response · Authors · 2024-11-26
>
> We thank reviewer XSHM for their comments.
>
> > I still do not understand what your definition of zero-shot coordination is, if you think that "mere state robustness" does not count as a solution to ZSC. And it's not just me, Reviewer GLBf also has this confusion, saying "So I wonder what is your definition of 'coordination'? Because I think generalizing to new partners is almost equal to ZSC."
>
> We follow the ZSC definition by Hu et al. The goal of the ZSC problem is to develop algorithms that enable agents to be trained independently while coordinating effectively at test time. There are multiple aspects that make ZSC challenging. State-coverage is one of them; however, it is not the only factor that makes coordination difficult. ZSC may also require learning grounded communication strategies or test-time protocols. Thus, while state-coverage contributes to the difficulty of the ZSC problem, solving state-coverage alone is not equivalent to solving ZSC.
>
>
> > One argument I am more sympathetic to is: the original Overcooked benchmark is almost saturated using a simple state robustness method. So, it is time to develop a harder benchmark. In this story, the state robustness algorithm is simply one algorithm for achieving ZSC, that works in the original Overcooked setting, but doesn't work for Overcookedv2.
>
> This is precisely the point. In Overcooked, we show that merely achieving state-coverage is sufficient to perform well in a ZSC setting. Consequently, it is not a good coordination benchmark, as state coverage represents only one aspect of ZSC. To address this limitation, we present a more challenging benchmark, OvercookedV2, which provides richer coordination challenges.
> While state-coverage remains part of the challenge in OvercookedV2, it is no longer sufficient to solve the benchmark. OvercookedV2 allows us to create scenarios that cannot be solved through state coverage alone. For instance, agents may need to rely on grounded communication or test-time protocol formation. This makes OvercookedV2 a much more comprehensive benchmark for evaluating ZSC, enabling the development of more complex and realistic scenarios.
> We will make this point clearer in the paper.
>
>
> > I'm not sure exactly what you are trying to say here, but I think it is probably untrue. There is no unique "best response" that is independent of the other agent, even in a fully observable setting. This was the main point in the original Overcooked paper (Carroll et al) -- Figure 1 demonstrates how incorrect beliefs about the partner agent's future actions can lead to bad outcomes.
>
> The reviewer is correct; our initial statement was misstated and as a result inaccurate. What we meant is this:
>
> In fully observable settings, each agent can independently compute all optimal joint-policies conditioned exclusively on the current state rather than on the action-observation history. It can then enact its part of one of the optimal joint-policies. If both agents do so, they will only fail to coordinate if they happen to choose joint-policies which are immediately incompatible. This can indeed happen, but we argue that it does so rarely in Overcooked, given our results in Section 5 which show that state augmentation greatly boosts XP even though we use non-recurrent policies.
>
> A follow-up [1] to Overcooked by some of the same authors echoes our claim, stating the following in Section 7:
> > "there is non-trivial strategically relevant information in Overcooked only in situations in which simultaneous decisions between incompatible optimal joint plans must be made. Given that such situations are rare, this provides a theoretical explanation as to why relatively good coordination with humans is achievable in this domain without any human data (Strouse et al., 2021). Our framework also explains why the introduction of simultaneous decisions (most easily through partial observability) can lead to benchmarks that are specifically more challenging for coordination, such as Hanabi"
>
> We hope this clarifies any confusion.
>
> > While movement can carry information (e.g., by moving toward a goal to indicate the intention of reaching it), it only conveys information if we assume the moving agent is approximately rational. A random agent performing the same movement would not communicate any bits of information.
> Yes, of course this depends on some minimal rationality assumptions, but this is also true of test-time protocol formation more generally. A random agent doesn't communicate information through its movement -- but you also can't form any test-time protocol with a random agent.
>
> This is correct; however, this is therefore not considered grounded communication. As stated previously, the test-time protocol formation layouts cannot be solved solely by relying on grounded communication. OvercookedV2 presents coordination challenges that include, but are not limited to, grounded communication.
>
> [1] Lauffer et al., 2023. https://arxiv.org/pdf/2306.09309

---

### Author Response · Authors · 2024-11-20

We thank all reviewers for the time taken to review the submission and for the constructive feedback. We address the common points raised by multiple reviewers in this general response.

We agree that additional baselines would be valuable and are currently running experiments with Fictitious Co-Play [1]. Implementing Off-belief learning (OBL) [2] is a complex effort beyond the scope of this paper. However, we will outline how we expect OBL to perform in the handcrafted coordination scenarios.

Moreover, we are including a more detailed explanation for the intended solution of our handcrafted scenarios. Additionally we are going to add a qualitative analysis of coordination failures in these scenarios and the corresponding GIFs.


[1] Strouse, D. J., et al. "Collaborating with humans without human data." Advances in Neural Information Processing Systems 34 (2021): 14502-14515.
[2] Hu, Hengyuan, et al. "Off-belief learning." International Conference on Machine Learning. PMLR, 2021.

---

### Meta-Review · Area_Chair_8hyz · 2024-12-20

**Metareview:**

This paper argues that OvercookedV1 which is a standard benchmark for ZSC is not sufficient because it is largely solved by state coverage. They argue that ZSC needs environments that can afford more than just this. The authors add features that induce partial observability, stochasticity, and asymmetric information across agents to form six new handcrafted coordination challenges. The authors show that this benchmark is not trivially solved by increasing state coverage.

Strengths:
Thoughtfully addresses some of the challenges in existing benchmarks for ZSC, and introduces a timely benchmark.
Provides a detailed analysis and empirical study

Weaknesses
The promised FCP experiment was not integrated as far as I know.
More baselines are needed, given it's a new benchmark paper.
A human evaluation would certainly be useful and add value

Overall I think that more baselines would help and the authors *really* need to add this baseline asap given that this was promised in the rebuttal, but this benchmark would add value to the ZSC community and offers a new and interesting challenge.

**Additional Comments On Reviewer Discussion:**

The reviewers were amazingly attentive and really engaged! Reviewer XSHM really engaged with the low level details and brought up the difference between ZSC and grounded communication. The authors had a good rebuttal to this and the reviewer raised their score accordingly. Similarly, the other reviewers asked for more baselines, and the authors promised to add FCP. As far as I can tell, this didn't happen. However, I am still voting to accept because of the benchmark being timely and very useful for the ZSC community even without this baseline. I really do need them to complete this baseline ASAP though.

---

### Decision · Program_Chairs · 2025-01-22

Accept (Poster)